

# Improving wind farm flow models by learning from operational data

Johannes Schreiber, Carlo L. Bottasso, Bastian Salbert, and Filippo Campagnolo

Wind Energy Institute, Technische Universität München, 85748 Garching bei München, Germany

**Correspondence:** Carlo L. Bottasso (carlo.bottasso@tum.de)

**Abstract.** This paper describes a method to improve and correct an engineering wind farm flow model by using operational data. Wind farm models represent an approximation of reality and therefore often lack accuracy and suffer from unmodeled physical effects. It is shown here that, by surgically inserting error terms in the model equations and learning the associated parameters from operational data, the performance of a baseline model can be improved significantly. Compared to a purely
data-driven approach, the resulting model encapsulates prior knowledge beyond the one contained in the training data set, which has a number of advantages. To assure a wide applicability of the method —including also to existing assets— learning is here purely driven by standard operational (SCADA) data. The proposed method is demonstrated first using a cluster of three scaled wind turbines operated in a boundary layer wind tunnel. Given that inflow, wakes and operational conditions can be precisely measured in the repeatable and controllable environment of the wind tunnel, this first application serves the
purpose of showing that the correct error terms can indeed be identified. Next, the method is applied to a real wind farm situated in a complex terrain environment. Here again learning from operational data is shown to improve the prediction capabilities of the baseline model.

## 1  Introduction

Knowledge of the flow at the rotor disk of each wind turbine in a wind power plant enables several applications, including wind
farm control, the provision of grid services, predictive maintenance, the estimation of life consumption, the feed-in to digital twins and power forecasting, among others.

This paper describes a new method to estimate turbine inflow within a wind farm. The main idea is to use an existing wind farm flow model to provide a baseline predictive capability; however, as all models contain approximations and may lack some physical phenomena, the baseline model is improved (or "augmented", which is the term used in this work) by adding
parametric correction terms. In turn, these extra elements of the model are learnt by using operational data. The correction terms capture effects that are typically not present in standard flow models (as, for example, secondary steering (Fleming et al., 2018) or wind farm blockage (Bleeg et al., 2018)), or that are highly dependent on a specific site or difficult to model upfront (as, for example, non-uniform inflow caused by local orography and vegetation).

Various wind farm flow models have been developed and are described in the literature. While Direct Numerical Simulation
(DNS) is still out of reach for practical applications due to its overwhelming computational cost, Large Eddy Simulation (LES) methods are now routinely used for the modeling of wind farm flows (Fleming et al., 2014; Breton et al., 2017). Although





invaluable for the understanding of the behavior of the atmospheric boundary layer and of wakes, LES is however still very expensive, so that its use outside of some specialized applications is limited. To reduce cost, one can resort to lower fidelity CFD models (Boersma et al., 2017), or to the extraction of reduced order models (ROMs) from higher fidelity ones (Bastine et al., 2014). Instead of deriving models from first principles, another widely adopted approach is to use engineering models,

which are expressed in the form of parametric analytical formulas with a limited number of degrees of freedom and hence a much reduced numerical complexity (Frandsen et al., 2006; Gebraad et al., 2014; Bastankhah and Porté-Agel, 2016). The present paper uses this last family of methods, although ideas similar to the ones developed here could be applicable also to higher fidelity models.

Even though engineering models are constantly improved and refined (Fleming et al., 2018), they will most likely always
exhibit only a limited accuracy in many practical applications, for example whenever an important role is played by effects such as orography, (seasonal) vegetation, spatial variability of the wind, sea state roughness, the erection of other neighbouring wind turbines, the presence of obstacles, and others. In addition, low fidelity models often lack some physics, e.g. the flow acceleration caused by wake and rotor blockage, secondary steering or others. The idea pursued in this paper is then to take a rather pragmatic approach: based on the realization that it will always be difficult —if not altogether impossible— to include
all effects and all physics in a model of limited numerical complexity, a given model is corrected by unknown parametric terms, which are then learnt by using operational data.

The idea of improving an existing model based on measurements is hardly new, and it is actually an important topic in the areas of controls and system identification. For example, in the field of wind farm flows, a Kalman filtering approach has been proposed by Doekemeijer et al. (2017) to update model predictions based on Lidar measurements. Here again the present
paper takes a more pragmatic approach, and model updating is based exclusively on data provided by the standard Supervisory Control And Data Acquisition (SCADA) systems that are typically available on contemporary wind turbines. On the one hand this has the advantage that the proposed method is applicable to existing assets, as it does not necessitate of extra sensors. On the other, given that stored SCADA data typically represents 10-minute averages, this also implies that the models obtained by this technique are of a steady-state nature. Although unsteady effects in wind farms are clearly important, steady-state models
are still very valuable and can support many of the applications listed above. In addition, nothing prevents the generalization of the proposed approach to unsteady flow models, assuming that the relevant higher frequency data sets are available, which is already the subject of ongoing work from these authors.

The contemporary literature —and not only in the field of wind energy— indicates an increasing interest in data-driven approaches. Just to give one single example related to wake modeling, a purely data-driven approach has been recently described
by Göçmen and Giebel (2018). However, the current enthusiasm for data should not make one forget that physics-based and analytical models are also extremely valuable because they often encapsulate significant knowledge on a given problem, often corroborated by a long experience. In fact, purely data-driven approaches suffer from a number of limitations that descend directly from a very simple and inevitable fact: a model that is exclusively based on data can only know what is contained in the data set that was used to build it. Typically, this means that a very significant amount of data is necessary to obtain a model
that is sufficiently general and accurate. Furthermore, the data has to cover the entire spectrum of operation of the system. This



also means that the model might have very poor knowledge (and hence poor performance) for rare situations or conditions that take place at the boundaries of the operating envelope, where few if any data points might be available.

An alternative to the purely data-driven approach is presented in this work, where a reference baseline model is augmented with parametric error terms, which are then identified using data. The baseline model already includes prior knowledge based on physics, empirical observations and experience. Therefore, even prior to the use of data, a minimum performance can be guaranteed. The model is augmented with parametric error terms, whose choice is driven by physics and the knowledge of the limitations of the baseline model. Once the errors are identified using operational data, their inspection can help clarify the causes of discrepancy between model and measurements. Eventually, this can be used to improve the underlying baseline model. Furthermore, by looking at the magnitude of the identified errors, significant deviations from the baseline model can be flagged to highlight issues with the model itself, the data or the training process.

Finally, it should be noted that the identification of the error terms can be combined with the tuning of the parameters of the baseline model. This addresses yet another problem: tuning the parameters of a model that lacks some physics may lead to unreasonable values for the parameters, as the model is "stretched" to represent phenomena that is does not contain. By the proposed hybrid approach, the simultaneous identification of the parameters of the baseline model together with the ones of the error terms eases this problem, as unmodeled phenomena can be captured by the model-augmenting terms, thereby reducing the chances of nonphysical tuning of the baseline parameters.

As for many identification problems, it is in general not possible to guarantee that all unknown parameters are observable and non-collinear given a set of measurements and, hence, given a certain informational content. To address this problem, the method proposed by Bottasso et al. (2014a) is used here, where the original unknown parameters are recast into a new set of statistically uncorrelated variables by using the Singular Value Decomposition (SVD) of the inverse Fisher information matrix. Once the problem has been solved in the space of the orthogonal uncorrelated parameters, the solution is mapped back into the original physical space. This approach not only avoids the ill-posedness of the original problem, but also allows one to clarify which physical parameters are visible given a certain data set.

The paper is organized as follows.

First, the baseline model is introduced, together with a detailed description of its proposed parametric corrections. Next, the new approach is applied to a cluster of scaled wind turbines operating in the atmospheric test section of the Politecnico di Milano wind tunnel (Bottasso et al., 2014b). Goal of this first application is that of showing that a correct identification of the error terms can be achieved. This is indeed possible in the controllable and repeatable conditions of a wind tunnel, where inflow and wake characteristics can be precisely measured, something that is hardly possible today in the field. Specifically, it is shown that the method can correctly learn the lack of uniformity of the wind tunnel inflow, which is akin to what happens in a real wind farm because of orographic effects. Similarly, it is shown that secondary steering, which is completely absent from the baseline model used here, can be learnt by using turbine power measurements only.

After having demonstrated the method in the known and controlled wind tunnel environment, a second application is developed that targets a real 43-turbine wind farm. Here results indicate that the augmented model has a markedly improved



prediction capability when compared to the baseline one, thanks primarily to the identification of orographic effects on the inflow and the tuning of other model parameters.

The paper is completed by two appendices, the first discussing the details of the SVD-based identification method, and the second reporting a more extended view of the results achieved in the wind tunnel.

## 2 Methods

### 2.1 Baseline wind farm flow model

The proposed method is applied here to the augmentation of the baseline wake model of Bastankhah and Porté-Agel (2016), implemented within the FLORIS framework (Doekemeijer et al., 2018).

By this model, given ambient wind conditions, steady state velocities within a wind farm can be computed, together with the corresponding operating states and power outputs of all its turbines. First, ambient conditions are estimated from un-waked machines operating in free stream, which in turn are identified by using the turbine yaw orientations and the wake model (Schreiber et al., 2018). Then, power and thrust of the upstream turbines are computed based on the turbine aerodynamic characteristics, alignment with the local wind direction and regulation strategy. Next, the wakes shed by these turbines are calculated in terms of their trajectory and speed deficit. In turn, this allows one to calculate the velocity at the rotor disks of the turbines immediately downstream. In case of multiple wake impingements on a rotor, a combination model is used to superimpose multiple wake deficits. Similarly, an added turbulence model is used to estimate the turbulence intensity at a downstream turbine rotor disk, as this local ambient parameter affects the expansion of the turbine wake. This process is repeated marching downstream throughout the wind farm until the last downstream turbine is reached.

In this work, the implementation uses the *selfSimilar* FLORIS velocity deficit model, the *rans* deflection model, the *quadraticRotorVelocity* wake combination model, and the *crespoHernandez* added turbulence model. The interested reader is referred to Bastankhah and Porté-Agel (2016), Crespo and Hernández (1996) and Doekemeijer et al. (2018) and references therein for detailed descriptions and derivations of these models.

Engineering wake models depend on a number of parameters, which should be tuned in order to obtain accurate predictions. For the specific model used in this work, these tunable factors are the wake parameters $\alpha$, $\beta$, $k_\mathrm{a}$, $k_\mathrm{b}$, $a_\mathrm{d}$, and $b_\mathrm{d}$, and the turbulence model parameters $TI_\mathrm{a}$, $TI_\mathrm{b}$, $TI_\mathrm{c}$, $TI_\mathrm{d}$ (Bastankhah and Porté-Agel, 2016).

In this work, the parameters are first set to an initial value, either taken from the literature or identified with ad hoc measurements. Corrections to the initial values are then expressed as

$$k = k^* + p_k, \tag{1}$$

where $k$ is a model parameter, $k^*$ its initial value and $p_k$ the correction. Although this is not strictly necessary, this redundant notation helps highlight the changes to the nominal model parameters obtained by the proposed procedure.





## 2.2 Model augmentation

The engineering model described earlier is a rather simple approximation of a flow through a wind power plant and it is therefore bound to have only a limited fidelity to reality, with a consequent only limited predictive accuracy. Even for more sophisticated future models, it is difficult to imagine that all relevant physics will ever be precisely accounted for. But even if
such a model existed, in practice one might simply not have all necessary detailed information on the relevant boundary and operating conditions that would be required. For example, one might not know with precision the conditions of the vegetation around and within a wind farm, with its effects on roughness and, hence, on the flow characteristics. In other words, it is safe to assume that all models are in error to some extent, and will probably always be.

To address this problem, the model can be pragmatically augmented with correction terms. Here one could take two alter-
native approaches: either a generic all-encompassing error term is added to the model, or "surgical" errors are introduced at ad-hoc locations in the model to target specific presumed deficiencies. The first approach could be treated with a brute-force parametric modeling approach, as for example by using a neural network. Here, the second approach was used, as it allows for more insight into the nature of the identified corrections. The specific parametric corrections used in the present paper are reviewed next. It is clear that these are only some of the many corrections that could be applied to the present baseline model,
so that the following does not pretend to be a comprehensive treatment of the topic. Nonetheless, results indicate that some of these corrections are indeed significant, and provide for a marked improvement of the baseline model.

**Non-uniform inflow.** The inflow to a wind farm can exhibit spatial variability, mostly because of orographic and local effects, especially in complex terrain conditions. For example, commercial wind resource assessment tools include topographic speed-up ratios customarily computed by CFD models (Jacobsen, 2019). In contrast to this established practice, no direct
or equivalent modeling of orographic effects are at present available in engineering wake models. Another reason for inflow variability may be due to wind farm blockage effects (Bleeg et al., 2018). Indeed, current wake models as the one used here assume that upstream turbines affect downstream ones through their wakes, but do not model the effects of downstream machines on the upstream ones. Depending on the wind direction and cross-wind location considered, the number and operating state of downstream turbines varies, which may induce a cross-wind speed variability in the
inflow.

To capture some of these effects, the model ambient flow speed $V_\infty$ is expressed here as a function of height above ground $Z$, cross-wind lateral position $Y$ and ambient wind direction $\Gamma$ as

$$V_\infty(Y, Z, \Gamma) = \left(1 + f_{\mathrm{augm,speed}}(Y, \Gamma, \boldsymbol{c}_{\mathrm{speed}}, \boldsymbol{p}_{\mathrm{speed}})\right) V_{\infty,0} \left(\frac{Z}{z_{\mathrm{h}}}\right)^{\alpha_{\mathrm{vs}}}, \qquad (2)$$

where $V_{\infty,0}$ is the reference (baseline uncorrected) ambient flow speed, and $z_{\mathrm{h}}$ the reference height of the vertically
sheared flow with exponent $\alpha_{\mathrm{vs}}$. Function $f_{\mathrm{augm,speed}}(Y, \Gamma, \boldsymbol{c}_{\mathrm{speed}}, \boldsymbol{p}_{\mathrm{speed}})$ is the speed correction term, modeled here with simple bilinear shape functions with node locations $\boldsymbol{c}_{\mathrm{speed}}$ and nodal values $\boldsymbol{p}_{\mathrm{speed}}$. Note that Eq. (2) could be extended to include also a longitudinal wind-aligned coordinate, similarly to the localized speed-up ratios of Jacobsen (2019). For simplicity, the present correction does not include the operating conditions of the downstream machines that,





in principle, would be necessary in order to more accurately model wind farm blockage effects. Therefore, the present correction can be interpreted as a primarily orography-induced one.

Local orographic effects and blockage may also induce variability of the wind direction $\Gamma$. Similarly, the vertical shear exponent $\alpha_{\mathrm{vs}}$ and turbulence intensity $I$ may vary, for example on account of non-uniform roughness induced by vegetation or other obstacles. To include these effects in the farm flow model, the baseline quantities are augmented as

$$\Gamma(Y) = \Gamma_{\mathrm{ref}} + Y f_{\mathrm{augm,dir}}(\Gamma_{\mathrm{ref}}, \boldsymbol{c}_{\mathrm{dir}}, \boldsymbol{p}_{\mathrm{dir}}), \tag{3a}$$

$$\alpha_{\mathrm{vs}}(\Gamma) = \alpha_{\mathrm{vs,ref}} + f_{\mathrm{augm,shear}}(\Gamma, \boldsymbol{c}_{\mathrm{shear}}, \boldsymbol{p}_{\mathrm{shear}}), \tag{3b}$$

$$I(\Gamma) = I_{\mathrm{ref}} + f_{\mathrm{augm,I}}(\Gamma, \boldsymbol{c}_{\mathrm{I}}, \boldsymbol{p}_{\mathrm{I}}). \tag{3c}$$

In all these expressions, $(\cdot)_{\mathrm{ref}}$ indicates a baseline reference quantity, while function $f_{\mathrm{augm,}(\cdot)}$ is a correction term based here on linear shape functions, with $\boldsymbol{c}_{(\cdot)}$ and $\boldsymbol{p}_{(\cdot)}$ the corresponding node locations and nodal values, respectively.

**Secondary steering.** By misaligning a wind turbine rotor with respect to the incoming flow direction, the rotor thrust force is tilted, thereby generating a cross-flow force that laterally deflects the wake. As shown with the help of numerical simulations by Fleming et al. (2018), this cross-flow force induces two counter rotating vortices that, combining with the wake swirl induced by the rotor torque, lead to a curled wake shape. As observed experimentally by Wang et al. (2018), the effects of these vortices result in additional lateral flow speed components, which are not limited to the wake itself but extend also outside of it. By this phenomenon, the flow direction within and around a deflected wake is tilted with respect to the upstream undisturbed direction. Therefore, when a turbine is operating within or close to a deflected wake, its own wake undergoes a change of trajectory —termed secondary steering— induced by the locally modified wind direction. Although models of this phenomenon are being developed (Martínez-Tossas et al., 2019), they significantly increase the computational cost and are not yet available in standard implementations of engineering wake models as the one used here.

The change of wind direction $\Delta\Gamma$ at a downstream turbine induced by secondary steering (indicated by the subscript ss) is modeled here as

$$\Delta\Gamma(y) = f_{\mathrm{augm,ss}}(Y - y_{\mathrm{wc}}, \Gamma_{\mathrm{init}}, \boldsymbol{p}_{\mathrm{ss}}), \tag{4}$$

where $f_{\mathrm{augm,ss}}$ is the correction term and $\tilde{y} = Y - y_{\mathrm{wc}}$ is the lateral distance to the wake centerline (see Fig. 1). According to the notation used in Eq. (6.12) of Bastankhah and Porté-Agel (2016), $\Gamma_{\mathrm{init}}$ indicates the initial wake direction of the closest upstream turbine. The correction term is expressed as the sum of two Gaussian functions, and more precisely

$$f_{\mathrm{augm,ss}}(\tilde{y}, \Gamma_{\mathrm{init}}, \boldsymbol{p}_{\mathrm{ss}}) =$$
$$\Gamma_{\mathrm{init}} \left( p_{\mathrm{ss,1}} \exp\left( -0.5 \Big( \frac{\tilde{y} + \mathrm{sgn}(\Gamma_{\mathrm{init}}) p_{\mathrm{ss,3}}}{p_{\mathrm{ss,2}}} \Big)^2 \right) - p_{\mathrm{ss,4}} \exp\left( -0.5 \Big( \frac{\tilde{y} + \mathrm{sgn}(\Gamma_{\mathrm{init}}) p_{\mathrm{ss,6}}}{p_{\mathrm{ss,5}}} \Big)^2 \right) \right), \tag{5}$$



where $\boldsymbol{p}_{\text{ss}} = (p_{\text{ss},1}, p_{\text{ss},2}, p_{\text{ss},3}, p_{\text{ss},4}, p_{\text{ss},5}, p_{\text{ss},6})$ is the vector of free parameters, where parameters 1 and 4 are related to the amplitude, 3 and 6 to the standard deviation, and 2 and 5 to the location of the correction functions. As the Gaussian functions are not centered at the wake centerline and the effect of secondary steering is assumed to be symmetric with respect to the misalignment angle, the correction term depends also on the direction of wake deflection $\text{sgn}(\Gamma_{\text{init}})$.

This particular choice of shape functions is motivated by the experimental results shown in Fig. 8b of Wang et al. (2018). Indeed, measurements reveal the presence of a lateral wake velocity whose maximum is displaced with respect to the wake centerline, as well as a slight lateral flow in the opposite direction that motivates the use of the second Gaussian function in the correction term introduced here.

Note that the change in local wind direction also leads to a slight lateral deflection of the non-uniform wind farm inflow
introduced previously. More precisely, for a turbine that is located $\Delta X$ behind an upstream turbine, the non-uniform inflow expressed by Eq. (2) is evaluated at $Y + \Delta X \sin(\Delta\Gamma)$ instead of $Y$.

The upper subplot of Fig. 1 shows the hub height flow speed for two wind turbines modeled in FLORIS, the turbine rotor disks being indicated with thick black lines. The wake centerlines and the undisturbed free stream wind direction are indicated by black dotted and dashed lines, respectively. The upstream turbine is misaligned with respect to the incoming
flow and therefore its wake is deflected laterally. Using the baseline wake model, the downstream turbine wake develops along the free stream wind direction. The lower subplot of the same figure shows the

effects of the secondary steering correction term presented above. The plot clearly shows that the downstream turbine wake path is affected by the locally changed wind direction.

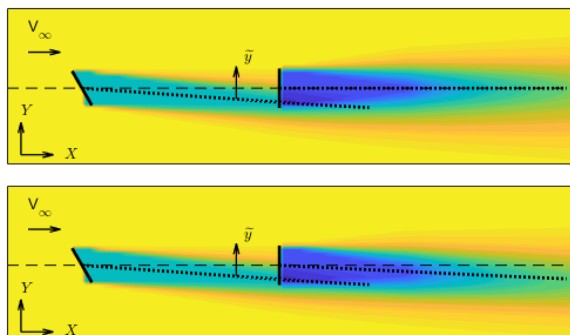

**Figure 1.** Effect of secondary steering on the trajectory of a downstream turbine. Top subplot: baseline wake model; lower subplot: baseline model augmented with the empirical correction term of Eq. (5).

**Non-Gaussian wake and flow acceleration.** Engineering wake models are based, among other hypotheses, on assumed shapes
of the speed deficit. For example, the present baseline model assumes a Gaussian distribution of the speed deficit within the wake. Another assumption is that the flow outside the wake is undisturbed, and equal to the free-stream. However,





these assumption can at times not be exactly satisfied, as already observed by Xie and Archer (2017) and Martínez-Tossas et al. (2019), among others. For example, aisle jets are local accelerations of the flow outside of the wake, produced by local blocking in the neighborhood of an operating turbine. It has been reported that aisle jets can induce local flow speedups in excess of $10\%$ of the undisturbed inflow (Dörenkämper et al., 2015).

To account for such effects, the wake velocity $V_{\mathrm{wake}}$ of the baseline model is corrected as

$$V_{\mathrm{wake}}(d_{\mathrm{wc}}) = V_{\mathrm{wake,FLORIS}}(d_{\mathrm{wc}})\Big(1 + f_{\mathrm{augm,acc}}(d_{\mathrm{wc}},\boldsymbol{c}_{\mathrm{acc}},\boldsymbol{p}_{\mathrm{acc}})\Big), \tag{6}$$

where $V_{\mathrm{wake,FLORIS}}$ is the baseline Gaussian wake speed profile, $d_{\mathrm{wc}}$ is the absolute distance to the wake center (which, at hub height, is equivalent to $|\tilde{y}|$), and $f_{\mathrm{augm,acc}}$ represents the correction term, modeled here as linear shape functions characterized by $\boldsymbol{c}_{\mathrm{acc}}$ node locations and $\boldsymbol{p}_{\mathrm{acc}}$ nodal values.

**Reduced power extraction due to non-uniform wind turbine inflow.** Numerical simulations conducted in FAST (Jonkman and Jonkman, 2018) using its Blade Element Momentum (BEM) implementation yielded a slight reduction in the rotor power coefficient for horizontally sheared flow, when compared to unsheared conditions with the same hub wind speed. Even though BEM can only give a rough indication for such effect, a correction of the power coefficient of the baseline model is introduced here in the form

$$C_{\mathrm{P}} = C_{\mathrm{P},\kappa=0}\Big(1 + p_{\kappa}\kappa^2\Big), \tag{7}$$

where $C_{\mathrm{P},\kappa=0}$ is the nominal power coefficient, $\kappa$ the equivalent horizontal linear shear coefficient on the rotor disk and $p_{\kappa}$ the free correction parameter. The linear shear $\kappa$ is either due to a lack of lateral uniformity of the inflow or to the impingement of a wake, and it is evaluated accordingly within the farm model.

**Wind speed dependent power loss in yaw misalignment.** The present baseline formulation models the power extraction of
a misaligned wind turbine using the cosine-law $C_{\mathrm{P}}(\gamma) = C_{\mathrm{P}}\cos(\gamma)^{p_{\mathrm{P}}}$, where $C_{\mathrm{P}}$ is the power coefficient of the wind-aligned turbine, $\gamma$ the misalignment angle with respect to local flow direction, and $p_{\mathrm{P}}$ the power loss exponent. Different values for the power loss exponent have been reported in the literature, ranging from the value of $1.4$ found by Fleming et al. (2017), to $1.8$ according to Schreiber et al. (2017), $1.9$ for Gebraad et al. (2015), all the way to the ideal value of $3$ that is expected if only the rotor-orthogonal ambient flow component contributes to power extraction (Boersma et al.,
2017). In addition, $p_{\mathrm{P}}$ might also depend on the regulation strategy used by the on-board controller. Here, the turbine power coefficient in misaligned operation is augmented as

$$C_{\mathrm{P}} = C_{\mathrm{P}}\cos(\gamma + p_{\mathrm{P0}})^{p_{\mathrm{P}} + p_{\mathrm{P,a}}(V - V_{\mathrm{rated}}) + p_{\mathrm{P,b}}}, \tag{8}$$

where $C_{\mathrm{P}}$ is the power coefficient of the flow-aligned turbine (possibly reduced by shear effects, as argued above), $p_{\mathrm{P0}}$ is the misalignment angle at which the turbine produces maximum power, while $V$ and $V_{\mathrm{rated}}$ are, respectively, the rotor
effective and rated wind speeds. Finally, $p_{\mathrm{P}}$ is the baseline exponent, while $p_{\mathrm{P,a}}$ and $p_{\mathrm{P,b}}$ are free parameters that model a linear wind speed dependency of the cosine law.



## 2.3 Parameter identification method

The parameters of the baseline model and of its corrections terms are identified with the method developed by Bottasso et al. (2014a). Details of the formulation are reported in Appendix A.

The formulation of the parameter estimation problem is independent on whether the parameters belong to the baseline model or to its correction factors. In this sense, one can use the same method to just tune the baseline parameters without considering the correction terms, just identify the correction terms at frozen baseline model, or identify concurrently both sets.

The formulation is based on the classical likelihood function, which describes the probability that a given set of noisy observations can be explained by a specific set of model parameters. By numerically maximizing this function, a set of parameters can be identified that most probably explains the measurements. Bound constraints are used to guide the process, and ensure convergence to meaningful results.

The accuracy with which the parameters can be estimated depends on how flat the likelihood function is with respect to changes in the parameters. For example, a flat maximum of the function implies that different nearby values of the model parameters are associated with similar values of the likelihood. These characteristics of the solution space are captured by the Fisher information matrix, which can be interpreted as a measure of the curvature of the likelihood function. Furthermore, it can be shown that the variance of the estimates is bound from below (Cramér-Rao bound) by the inverse of the Fisher matrix (Jategaonkar, 2015). Although the analysis of the Fisher information is useful for the understanding of the well-posedness of an estimation problem and of the quality of the identified model, it does not offer a constructive way of reformulating a given ill-posed problem. Indeed, a flat solution space and collinear parameters are to be expected in the present case, given the complex couplings and dependencies that may exist among the various parameters of a wind farm flow model and its correction terms.

To overcome this limitation of the classical maximum likelihood formulation, following Bottasso et al. (2014a), the original physical parameters of the model are transformed into an orthogonal parameter space, by diagonalizing the Fisher matrix using the SVD. This way, as the parameters are now statistically decoupled, one can set a lower observability threshold, and retain in the analysis only the ones that are in fact observable given the available set of measurements. Once the problem is solved, the uncorrelated parameters are mapped back into the original physical space.

As shown later on, this approach achieves multiple goals: it allows one to successfully solve a maximization problem with many free parameters, some of which might be interdependent on one other or not observable in a given data set; it reduces the problem size, retaining only the orthogonal parameters that are indeed observable; it highlights, through the singular vectors, the interdependencies that may exist among some parameters of the model, which provides for a useful interpretation tool that may guide the reformulation of parts of the model and its correction terms.

## 3 Results

The results section is split into two parts: the application of the method to wind tunnel measurements, reported in Section 3.1, and to a real wind farm consisting of 43 wind turbines, described in Section 3.2. The former aims at a verification of the



correctness of the identified augmentations, given the known and controllable conditions of the scaled experiments, while the latter is meant to offer a first glimpse on the practical applicability of the new method in the field.

## 3.1 Wind tunnel verification

Whether some identified model corrections are indeed physical or only an artefact of the model-measurement mismatch is
difficult to prove in general. From this point of view, wind tunnel experiments provide for a unique opportunity to verify the concept proposed in this paper. Indeed, the overall flow within a cluster of turbines can be measured with good accuracy, and the experiments can be repeated in multiple desired operating conditions. The aim of this section is then to show that, even in the presence of multiple possibly overlapping model terms, the correct improvements to a baseline model can be learnt from operational data only.

### 3.1.1 Experimental setup

The experimental setup is composed by a scaled cluster of three G1 wind turbines, each of them equipped with active yaw, pitch and torque control. The turbines were operated in the boundary layer test section of the wind tunnel of the Politecnico di Milano. Details on the models and the wind tunnel are reported, among other publications, in Campagnolo et al. (2016a, b, c).

The turbines are labelled WT1, WT2 and WT3, starting from the most upstream one and moving downstream. The machines
are mounted on a turntable, which allows for changing the wind direction with respect to the wind farm layout. In the nominal configuration, i.e. for a turntable rotation $\gamma_{\mathrm{TT}} = 0°$, the three turbines are aligned with the wind tunnel main axis —and hence with the flow velocity vector. The turbines are installed with a longitudinal spacing of 5 diameters (D), as shown in the left part of Fig. 2 with a view looking down towards the wind tunnel floor. As indicated in the figure, positive turntable rotations are clockwise. For $\gamma_{\mathrm{TT}} \neq 0°$, the longitudinal distance between the turbines decreases slightly. However, considering that in this
work the largest investigated turntable angle was $\pm 11.5°$, the longitudinal distance varied only between $4.9D$ and $5D$.

A pitot probe was placed at turbine hub height, 3D upstream of the first G1 in the nominal configuration. The probe was therefore not placed on the turntable, and its position remained fixed with respect to the wind tunnel test section. A wind-tunnel-fixed reference frame, used in the following to discuss the results, is also depicted in Fig. 2. Its origin is placed at the turntable center, while the frame $X$ axis is aligned with the wind direction, the $Y$ axis points left looking downstream, and
hence $Z$ points vertically up from the floor to complete a right handed triad.

The yaw angle $\gamma_{\mathrm{WT}i}$ of the $i$-th wind turbine is positive for a counterclockwise rotation looking down onto the floor, as shown for WT1 in Fig. 2, and null when the rotor disk is orthogonal to $X$ and, therefore, to the nominal wind direction.

The right part of Fig. 3 shows a photo of the cluster of turbines, looking downstream with WT1 in the foreground. The wind tunnel floor is of a blue color, whereas the turntable is black.
The ambient wind speed $V_{\infty,0}$ measured by the pitot tube was, for all conducted experiments, between 5.20 and 5.75 m/s, which correspond to slightly below-rated conditions. The ambient turbulence intensity was equal to $6.12\%$, while the vertical shear was $\alpha_{\mathrm{vs}} = 0.144$.





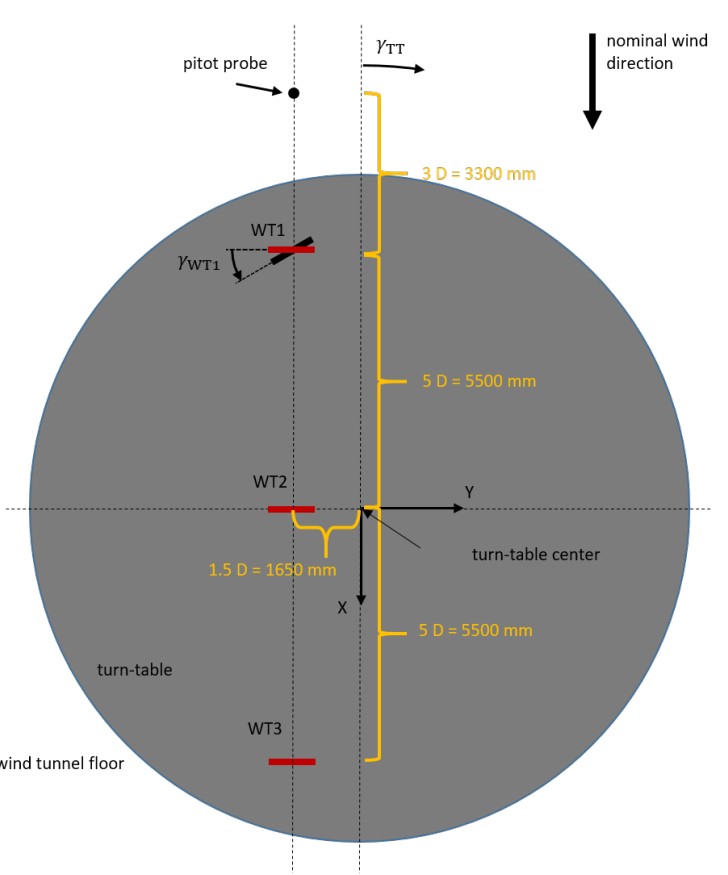

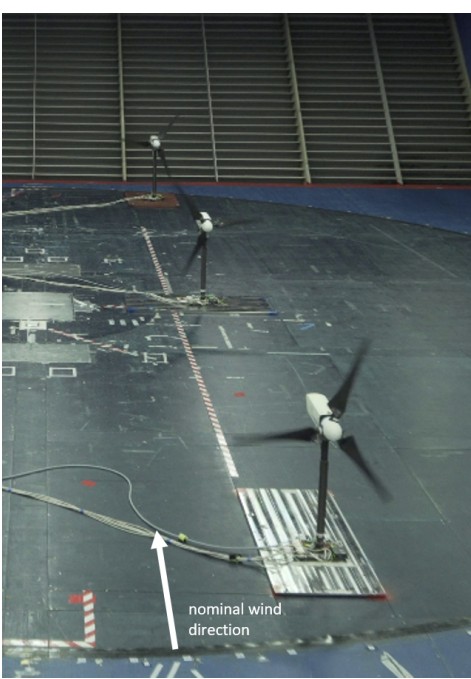

**Figure 2.** Wind farm layout for a null turntable rotation, looking down onto the wind tunnel floor.

**Figure 3.** View looking downstream of the cluster of three G1 turbines.

### 3.1.2 Model setup

The FLORIS model implementation used in this work is the one available online (Doekemeijer and Storm, 2018). All the baseline model parameters were first identified based on single turbine wake measurements (Campagnolo et al., 2019), and their values are reported in Table 1.

**Table 1.** Initial FLORIS parameters for the G1 turbine.

| $\alpha^*$ | $\beta^*$ | $k_{\mathrm{a}}^*$ | $k_{\mathrm{b}}^*$ | $a_{\mathrm{d}}^*$ | $b_{\mathrm{d}}^*$ | $TI_{\mathrm{a}}^*$ | $TI_{\mathrm{b}}^*$ | $TI_{\mathrm{c}}^*$ | $TI_{\mathrm{d}}^*$ |
|---|---|---|---|---|---|---|---|---|---|
| 0.9523 | 0.2617 | 0.0892 | 0.027 | 0 | 0 | 0.082 | 0.608 | $-0.551$ | $-0.2773$ |

5    Figure 4 shows the G1 power $C_\mathrm{P}$ and thrust $C_\mathrm{T}$ coefficients as functions of wind speed $V$. The curves were obtained from dynamic simulations conducted in turbulent inflow, using the same controllers implemented on the scaled models. The $C_\mathrm{P}$ and $C_\mathrm{T}$ vs. tip-speed-ratio (TSR) and blade pitch setting curves were obtained based on a BEM formulation using experimentally-tuned

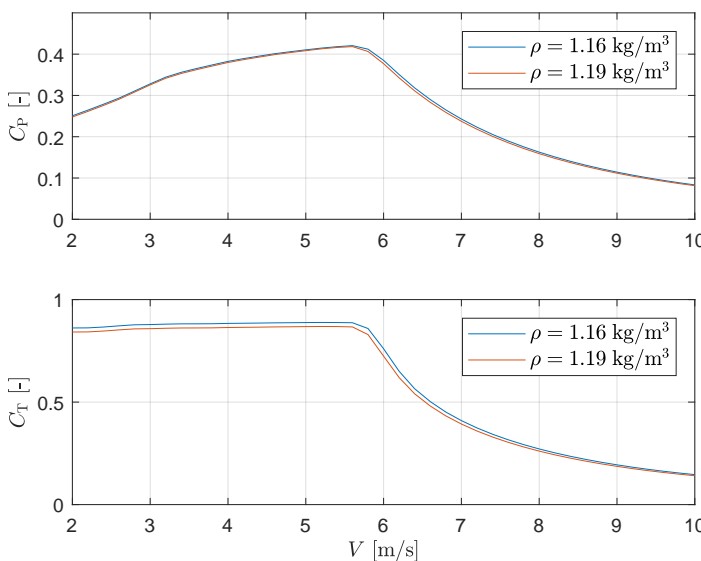

**Figure 4.** Power and thrust coefficients vs. wind speed for the G1 turbine.

airfoil polars (Bottasso et al., 2014a). As the turbine controller does not consider variations of air density $\rho$, the coefficients shown in the figure exhibit a slight dependency on this ambient parameter. Within FLORIS, this effect is taken into account by interpolating between the coefficients based on the actual density measured in the wind tunnel during each experiment. For all reported test conditions, air density varied in the range $\rho \in [1.159, 1.185]$ kg/m$^3$. The power loss exponent in misaligned

conditions was evaluated experimentally to be $p_\mathrm{P} = 2.1741$, while for thrust the coefficient was found to be $p_\mathrm{T} = 1.4248$.

The ambient wind speed was determined from the pitot tube. It was observed that, by using this value, the power of a free-stream turbine predicted by the FLORIS model was slightly underestimated, most probably due to the sheared flow. To correct for this effect, measurements provided by the pitot tube were scaled by the factor $1.0176$, which was computed in order to match simulated and measured power. Furthermore, in the original FLORIS implementation the power of a turbine

is computed as $P = 1/2\rho A V_\mathrm{avg}^3 C_\mathrm{P}$, where $V_\mathrm{avg}$ is the average wind speed at the rotor disk and $A$ the rotor disk area. Here, power was computed by integrating over the rotor disk area, i.e. $P = 1/2\rho \int_A V^3 C_\mathrm{P} \mathrm{d}A$, which is believed to be slightly more accurate even though it involves a minor increase in computational effort.

### 3.1.3   Ranking of correction terms

To initially assess the role of the various parameters, a ranking analysis was conducted. The parameters were clustered in sets,

depending on their role in the model. A first identification was performed using all parameter sets, yielding the presumed best value, denoted as $J_\mathrm{ref}$, of the cost function expressed by Eq. (A6). The analysis was then repeated multiple times, each time removing one parameter set from the optimization. By looking at the resulting change in the value of the cost function, one may





then rank the various parameter sets in order of importance. The analysis is based on a total of 190 experimental observations, as described in greater detail in the following subsection.

All augmentation terms described in Section 2.2 were considered, except for the lateral variation in wind direction and the wind direction dependent vertical shear, as they are not applicable to the wind tunnel experiments. The non-uniform flow speed was modeled using five nodes located at $\boldsymbol{c}_{\text{speed}}(Y) = [-3, -2, -1, 0, 1]$ m. As only the turbine positions with respect to the flow is altered by using the turntable, a wind direction dependency was not included in this correction term. For the non-uniform inflow and secondary steering augmentations, the parameter initial values, lower and upper bounds and definitions are shown in Table 2.

**Table 2.** Definition of the parameters, together with their initial values, lower and upper bounds, and identified values.

| $i$ | $p_i$ | $p_{\text{lb},i}$ | $p_{\text{ub},i}$ | $p_{\text{init},i}$ | $p_{\text{opt},i}$ | Implementation |
|---|---|---|---|---|---|---|
| $1-5$ | $\boldsymbol{p}_{\text{speed}}$ | $-0.1$ | $0.1$ | $0$ | $[0.079, 0.029, ...$ | $f_{\text{augm,speed}}(Y, Z, 0, \boldsymbol{c}_{\text{speed}}, \boldsymbol{p}_{\text{speed}})$ |
| | | | | | $-0.051, -0.006, 0]$ | $\boldsymbol{c}_{\text{speed}} = [-3, -2, -1, 0, 1]$ m |
| $6-11$ | $\boldsymbol{p}_{\text{ss}}$ | $[-3, 0, ...$ | $[3, 1.5, ...$ | $[-0.5, 0.5, ...$ | $[-0.94, 0.63, ...$ | $f_{\text{augm,ss}}(\tilde{y}, \Gamma_{\text{init}}, \boldsymbol{p}_{\text{ss}})$ |
| | | $-3, -3, ...$ | $3, 3, ...$ | $0.2, -0.25, ...$ | $0.20, -0.48, ...$ | |
| | | $0, -3]$ | $1.5, 3]$ | $0.5, -0.2]$ | $0.73, -0.28]$ | |

Figure 5 shows the relative increase of the cost function when eliminating one parameter set at a time. The figure clearly indicates that the most important parameters are the ones modeling laterally non-uniform speed and secondary steering. Indeed, this particular wind tunnel, due to its internal configuration and large width, does present a significant non-uniform flow speed, as already discussed by Campagnolo et al. (2019). Likewise, the effect of secondary steering is particularly important and should not be neglected for accurate predictions in misaligned conditions, as already reported in various publications. Based on these results, in the following only non-uniform inflow and secondary steering corrections are considered.

### 3.1.4 Results

A total of 451 observations were available, including 11 different turntable positions and thus wind farm layouts, with turbine yaw misalignments ranging from $-40°$ to $+40°$. A total of 190 observations were used to identify the 5 parameters associated with non-uniform inflow speed and the 6 associated with secondary steering, whereas the remaining data points were used for model validation. The various tested configurations in terms of turbine misalignments and turntable positions are reported in the figures of Appendix B.

Among all the available measurements gathered at each operating condition, only the steady-state power of the wind turbines was utilized. The model outputs $\boldsymbol{y}$ (see Eq. (A1)) are defined as

$$\boldsymbol{y} = \frac{1}{P_{\text{ref}}} \begin{bmatrix} P_{\text{WT1}} \\ P_{\text{WT2}} \\ P_{\text{WT3}} \end{bmatrix}, \tag{9}$$

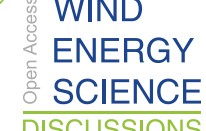

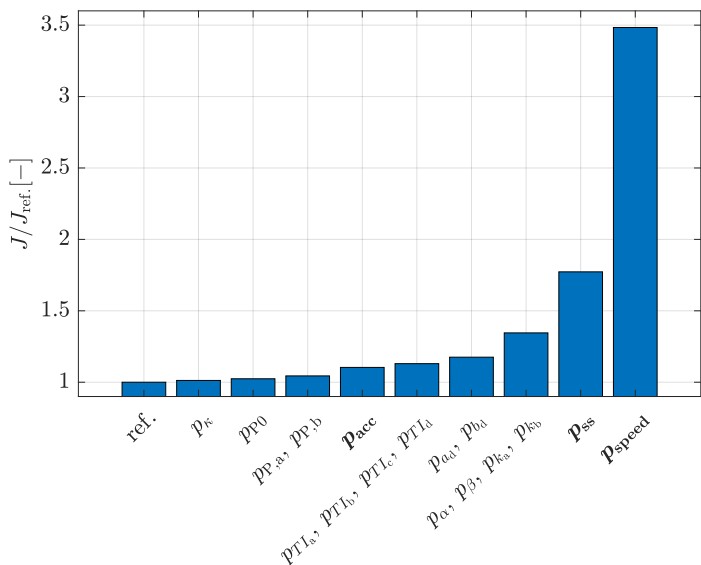

**Figure 5.** Relative increase of the optimization cost function when eliminating one parameter set at a time.

where $P_{\mathrm{WT}i}$ is the power of the $i$-th wind turbine and $P_{\mathrm{ref}} = 37.6$ W is a reference value used as scaling factor. Based on experience, a diagonal measurement noise covariance matrix $\boldsymbol{R}$ with all three terms equal to $\sigma^2 = 0.025^2$ was specified.

The threshold of the highest acceptable standard variance $\sigma_t^2$ for the orthogonal parameters was set to $0.01$. As the parameters are scaled within a range $[-1, 1]$, the threshold corresponds to a relative variance of $2\%$. Wind-aligned operating condition

(i.e., $\gamma_{\mathrm{WT1}} = \gamma_{\mathrm{WT2}} = \gamma_{\mathrm{WT3}} = 0°$) were weighted with a factor of 2, to increased their importance in the parameter estimation process.

The constrained optimization problem (A5) was solved in Matlab using the *fmincon* function with the *interior-point* algorithm (Mathworks, 2019). As the baseline model with its initial nominal values ($\boldsymbol{p} = \boldsymbol{p_{\mathrm{init}}}$) is far away from the optimal solution, a first optimization was performed including only the inflow correction. Afterwards, three iterations were conducted

including all 11 parameters. At each iteration, a total of 8 orthogonal parameters could be identified within the specified variance threshold. The method converged very quickly, as the identified parameters and the residual did not change significantly after the first iteration. Figure 6 shows on the left the initial variance of all 11 orthogonal parameters, and on the right the variance computed after the first iteration. The horizontal black line indicates the threshold $\sigma_t^2$.

Interestingly, the 11-th orthogonal parameter seems to have a very low observability. Table 3 shows the transformation matrix

$\boldsymbol{V}^T$ that links the physical parameters to the orthogonal ones ($\boldsymbol{\Theta} = \boldsymbol{V}^T\boldsymbol{p}$, see Eq.( A14)). The 11-th orthogonal parameter is almost entirely associated with $p_{\mathrm{speed},5}$, which corresponds to the inflow speed augmentation node at position $Y = 1$ m. Indeed, the location of this node is such that it has only a very marginal effect on the turbine outputs and, hence, a very low observability, as shown later on in Fig. 7. The transformation matrix reported in Table 3 also shows that the other two orthogonal parameters





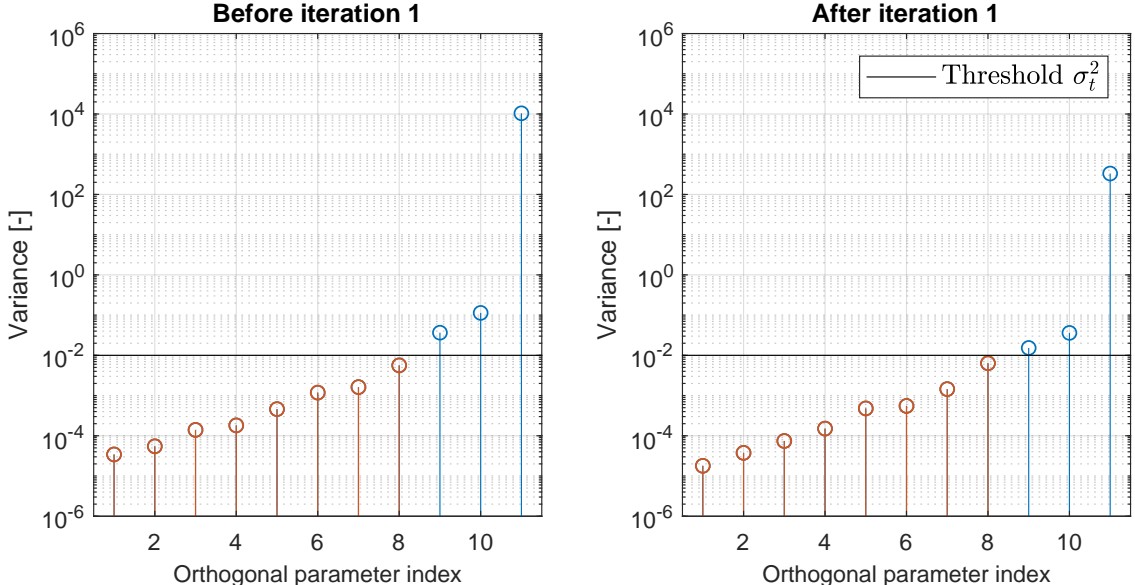

**Figure 6.** Variance of the orthogonal parameters before (left) and after (right) the first iteration. The identifiable orthogonal parameters are shown in red, whereas all others are shown in blue.

with low observability (9 and 10) represent secondary steering modes, mainly associated with the second Gaussian function of the correction term.

Table 4 presents the correlation matrix $\Psi$ (cf. Eq. (A9)), and shows a clear and to be expected dependency among neighbouring inflow parameters. Among the secondary steering parameters, strong but less obvious correlations are present, which

suggest that a simplification of the assumed correction term might be possible.

Figure 7 shows the identified inflow augmentation function. In the picture, whiskers indicate the parameter uncertainty $\sigma_i$, computed based on the Cramér-Rao lower error bound as $\boldsymbol{\sigma} = \sqrt{\mathrm{diag}(\boldsymbol{P})}$ (cf. Eq. (A8)). The same figure reports also measurements obtained with hot wire probes in the empty wind tunnel at three different heights above the floor. These measurements, and especially the ones at hub height, are in good agreement with the estimates provided by the proposed method. The figure

also reports (with $\times$ symbols) the lateral position of the upstream turbine for the investigated turntable rotations. Noting that all points are shifted to the left, helps explain why the parameter associated with the inflow node at $Y = 1$ m has a very low —but still finite— observability.

The identified secondary steering augmentation term is visualized in Fig. 8. The plot shows the wind direction change $\Delta\Gamma$ as a function of the distance $\tilde{y}$ to the wake centerline for a turbine misalignment of $20°$. The gray shaded area shows the

15 uncertainty band $p_{\mathrm{opt},i} \pm \sigma_i$. Consistently with the findings of Wang et al. (2018), the maximum change in wind direction is found at approximatively $0.3$ D on the leeward side of a deflected wake. The maximum magnitude of secondary steering in this operating condition is

$1.9°$, which is is again comparable to the results of Wang et al. (2018).





**Table 3.** Transformation matrix $\boldsymbol{V}^T$ after the first iteration. Each row corresponds to a different orthogonal parameter.

| | $p_{\text{speed},1}$ | $p_{\text{speed},2}$ | $p_{\text{speed},3}$ | $p_{\text{speed},4}$ | $p_{\text{speed},5}$ | $p_{\text{ss},1}$ | $p_{\text{ss},2}$ | $p_{\text{ss},3}$ | $p_{\text{ss},4}$ | $p_{\text{ss},5}$ | $p_{\text{ss},6}$ |
|---|---|---|---|---|---|---|---|---|---|---|---|
| 1 | -0.0 | 0.0 | 0.0 | -0.0 | -0.0 | -0.7 | 0.2 | -0.0 | 0.7 | -0.1 | -0.1 |
| 2 | -0.2 | -0.4 | -0.3 | -0.1 | -0.0 | 0.2 | -0.1 | -0.7 | 0.3 | 0.1 | 0.3 |
| 3 | 0.0 | -0.6 | -0.6 | -0.1 | 0.0 | -0.1 | 0.0 | 0.4 | -0.1 | -0.0 | -0.2 |
| 4 | -0.4 | -0.6 | 0.6 | 0.3 | 0.0 | -0.0 | 0.0 | 0.1 | -0.0 | -0.0 | -0.0 |
| 5 | -0.7 | 0.2 | -0.1 | -0.2 | -0.0 | 0.2 | 0.5 | 0.2 | 0.1 | -0.1 | 0.1 |
| 6 | -0.5 | 0.2 | -0.1 | -0.1 | 0.0 | -0.4 | -0.7 | -0.0 | -0.2 | 0.1 | -0.2 |
| 7 | 0.1 | -0.2 | 0.3 | -0.9 | -0.0 | -0.0 | -0.1 | 0.1 | -0.0 | -0.0 | 0.1 |
| 8 | 0.0 | 0.0 | -0.0 | 0.1 | -0.0 | 0.3 | -0.5 | 0.5 | 0.5 | 0.1 | 0.4 |
| 9 | -0.1 | 0.0 | 0.0 | -0.1 | 0.0 | 0.2 | 0.1 | 0.0 | 0.2 | 0.8 | -0.5 |
| 10 | 0.0 | -0.0 | 0.0 | -0.0 | -0.0 | 0.4 | -0.2 | -0.1 | 0.3 | -0.6 | -0.6 |
| 11 | 0.0 | -0.0 | 0.0 | -0.0 | 1.0 | -0.0 | -0.0 | 0.0 | -0.0 | -0.0 | 0.0 |

**Table 4.** Correlation coefficients $\Psi$ after the first iteration.

| | $p_{\text{speed},1}$ | $p_{\text{speed},2}$ | $p_{\text{speed},3}$ | $p_{\text{speed},4}$ | $p_{\text{speed},5}$ | $p_{\text{ss},1}$ | $p_{\text{ss},2}$ | $p_{\text{ss},3}$ | $p_{\text{ss},4}$ | $p_{\text{ss},5}$ | $p_{\text{ss},6}$ |
|---|---|---|---|---|---|---|---|---|---|---|---|
| $p_{\text{speed},1}$ | 1.0 | -0.5 | 0.2 | -0.1 | 0.2 | -0.1 | -0.1 | -0.0 | -0.1 | -0.2 | 0.2 |
| $p_{\text{speed},2}$ | -0.5 | 1.0 | -0.7 | 0.5 | -0.2 | -0.0 | 0.0 | 0.1 | 0.0 | 0.2 | 0.0 |
| $p_{\text{speed},3}$ | 0.2 | -0.7 | 1.0 | -0.7 | 0.2 | 0.1 | -0.0 | -0.1 | 0.1 | -0.1 | -0.2 |
| $p_{\text{speed},4}$ | -0.1 | 0.5 | -0.7 | 1.0 | -0.4 | -0.1 | -0.0 | 0.1 | -0.0 | 0.1 | 0.2 |
| $p_{\text{speed},5}$ | 0.2 | -0.2 | 0.2 | -0.4 | 1.0 | -0.1 | -0.1 | 0.0 | -0.1 | -0.3 | 0.2 |
| $p_{\text{ss},1}$ | -0.1 | -0.0 | 0.1 | -0.1 | -0.1 | 1.0 | -0.6 | -0.1 | 0.9 | -0.4 | -0.8 |
| $p_{\text{ss},2}$ | -0.1 | 0.0 | -0.0 | -0.0 | -0.1 | -0.6 | 1.0 | -0.3 | -0.7 | 0.6 | 0.3 |
| $p_{\text{ss},3}$ | -0.0 | 0.1 | -0.1 | 0.1 | 0.0 | -0.1 | -0.3 | 1.0 | 0.2 | 0.4 | 0.6 |
| $p_{\text{ss},4}$ | -0.1 | 0.0 | 0.1 | -0.0 | -0.1 | 0.9 | -0.7 | 0.2 | 1.0 | -0.2 | -0.6 |
| $p_{\text{ss},5}$ | -0.2 | 0.2 | -0.1 | 0.1 | -0.3 | -0.4 | 0.6 | 0.4 | -0.2 | 1.0 | 0.3 |
| $p_{\text{ss},6}$ | 0.2 | 0.0 | -0.2 | 0.2 | 0.2 | -0.8 | 0.3 | 0.6 | -0.6 | 0.3 | 1.0 |



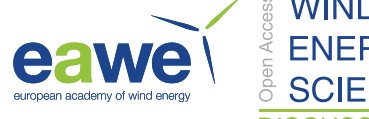

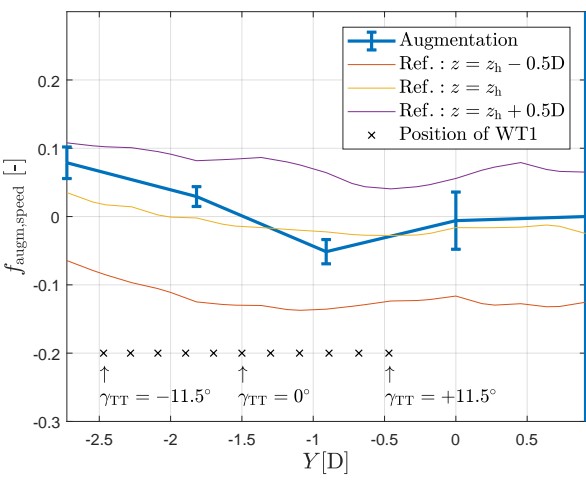

**Figure 7.** Identified non-uniform inflow speed augmentation term (solid line) and associated standard deviation (whiskers). Hot wire measurements at different heights above the floor are shown in thin solid lines. The upstream turbine (WT1) position is shown by × markers for all investigated turntable positions.

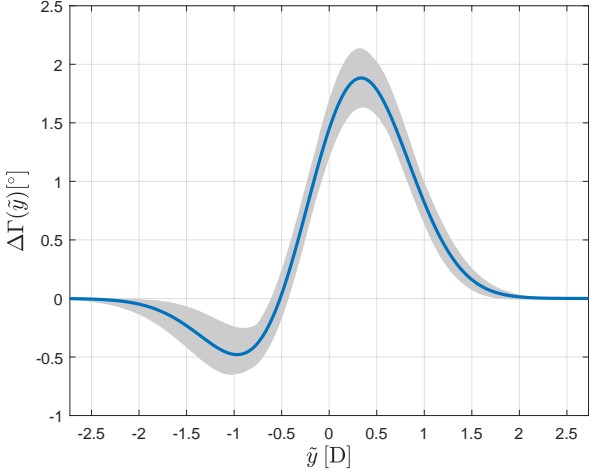

**Figure 8.** Identified wind direction change $\Delta\Gamma$ due to secondary steering as a function of distance $\tilde{y}$ to the wake centerline for a turbine misalignment of $20°$. The grey shaded area shows the uncertainty band.





The validity of the augmentation terms, identified as explained, was assessed by comparing the results of the simulation model with experimental wake measurements from a different test campaign. The setup was identical to the one considered here, except for the fact that only the first two upstream wind turbines were installed in the wind tunnel. At the downstream distance where the third wind turbine should have been installed, flow velocity measurements were obtained at turbine hub

height using hot wire probes. Figure 9 shows wake profiles for the turntable position $\gamma_{\text{TT}} = 0°$ for various combinations of turbine yaw misalignments, as indicated by the subplot titles. Each subplot is accompanied by two flow visualizations, one based on the baseline FLORIS model and the other on its augmented version. The figures also include the points at which the flow was measured with the probes.

In the left subplots, the improvements of the augmented model with respect to the baseline FLORIS are exclusively due to

the inflow correction, as the upstream turbine is aligned with the flow, and therefore there are no secondary steering effects. In the right subplots, the upstream turbine is misaligned ($\gamma_{\text{WT1}} = 30°$) and secondary steering effects are present. Taking into account that model augmentation was obtained exclusively by turbine power measurements, the improved matching of the wake profiles is remarkable. Still, even with the extra correction terms some model mismatches are present; these might be caused by the wake combination model, which was not augmented in this study.

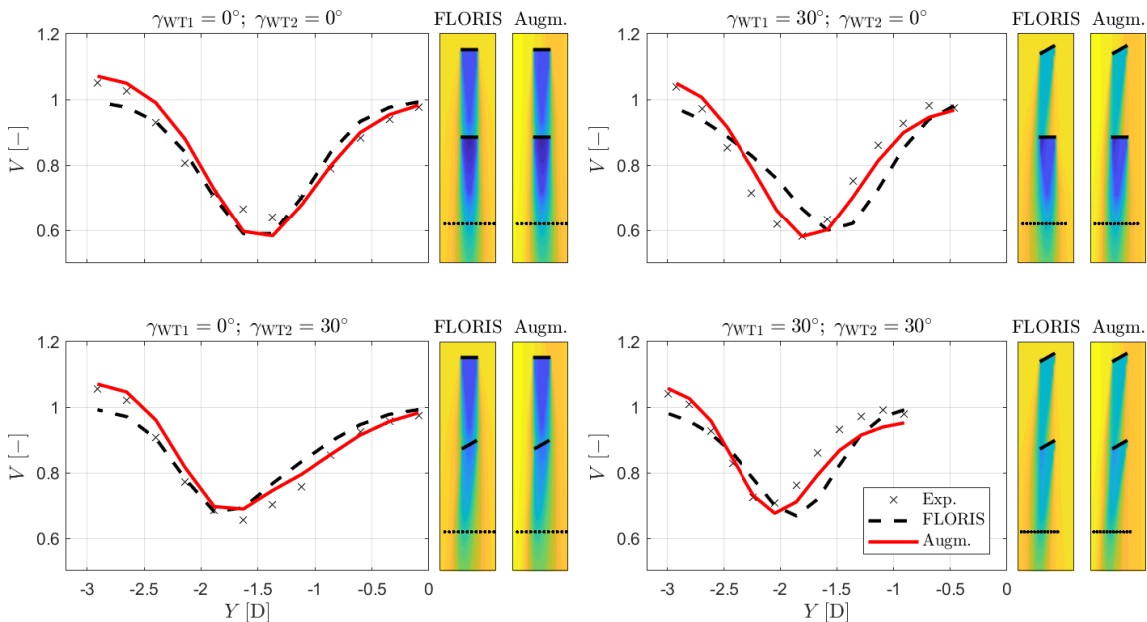

**Figure 9.** Wake profiles 5D behind WT2 for various combinations of turbine yaw misalignment. Experimental values are indicated by the × symbols. Each subplot is accompanied by two flow visualizations based on the FLORIS model and its augmented version.

The turbine power coefficients are computed as

$$C_{\text{P,i}} = \frac{P_{\text{WT}i}}{0.5\rho A V_{\infty}(Y_{\text{WT}i}, z_{\text{h}}, 0)^3}, \qquad (10)$$



where $V_\infty$ is the augmented inflow function given by Eq. (2), evaluated at the respective turbine position $Y_{\mathrm{WT}i}$ and hub height $z_h$. A detailed overview of the results is offered by the figures of Appendix B, which report the power outputs and the model errors for all wind farm configurations. For readability, here a more synthetic overview of the results is presented, by condensing the information contained in Fig. B1, B2 and B3 in the probability density plots of Fig. 10. This figure shows the results for the baseline FLORIS model using a black dashed line, for the 11-parameter augmented model (i.e., only non-uniform inflow speed and secondary steering corrections) using a red solid line, and the 27-parameter augmented model (i.e., including all additional augmentation terms presented earlier on) using a red dotted line. The root mean squared errors $\epsilon_{\mathrm{RMS}}$ are shown in the respective legends.

Note that the FLORIS error distribution shows two peaks for WT1 and WT3, indicating the presence of two uncorrelated errors. The 11-parameter model removes these peaks, even though a smaller pair of peaks remains for WT2 and WT3, indicating additional errors that only the 27-parameter augmented model is able to capture.

Here again the trend is clear: the addition of non-uniform speed and secondary steering increases substantially the accuracy of the baseline model, with additional small —but not insignificant— gains offered by the additional correction terms. Finally, there is still room for improvement, possibly through extra correction terms not yet explored.

## 3.2 Field application

In this section the model augmentation and identification method is applied to a full scale wind farm, to test its applicability and usability in a realistic scenario. In such conditions, it is often difficult to assess weather the identified model corrections are indeed physical or not, due to a lack of knowledge of the actual ground truth. To deal with this problem, the classical approach of splitting the data set was used here: first, a relatively small subset of measurements is used for model and error identification; then, the rest of the data set is used for a verification of the generality of the identified model, and of its improved performance with respect to the baseline one.

### 3.2.1 Wind farm and data pre-processing

The onshore wind farm is situated close to Sedini, on the Italian island of Sardinia, and it consists of 43 GE1.5s and GE1.5sle wind turbines, as specified in Table 5.

**Table 5.** Turbine specifications

| Type | Rated power [MW] | Cut-in wind speed [m/s] | Rated wind speed [m/s] | Rotor diameter [m] | Hub height [m] | Installed units [-] |
|---|---|---|---|---|---|---|
| GE1.5s | 1.5 | 4 | 13 | 70.5 | 65 | 36 |
| GE1.5sle | 1.5 | 3.5 | 12 | 77 | 80 | 7 |



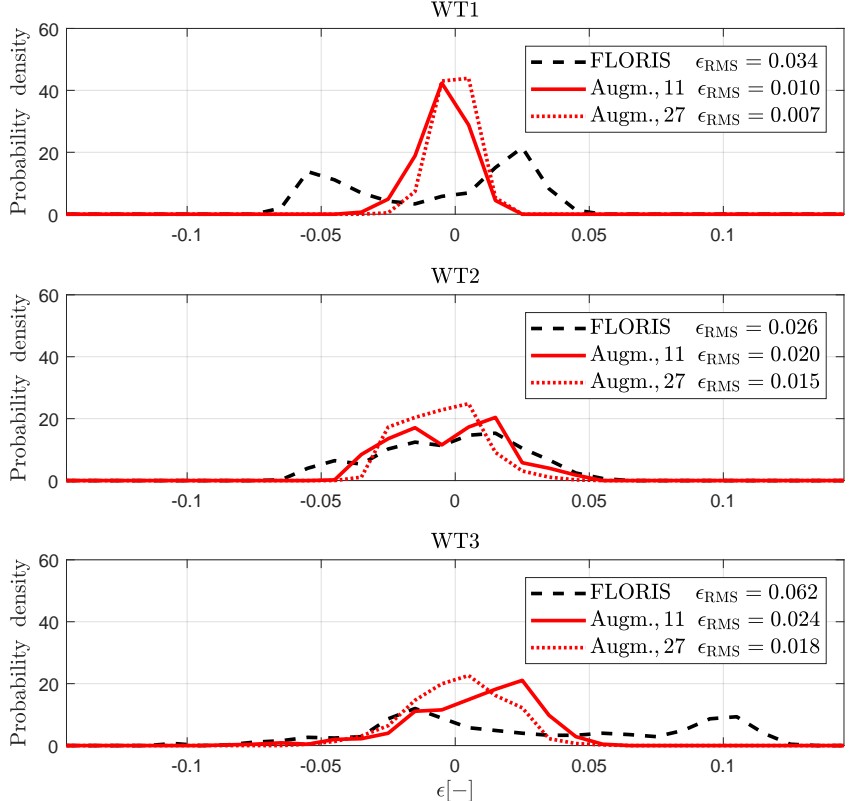

**Figure 10.** Error distributions for each turbine for all tested configurations, for the baseline FLORIS model (black dashed line), the 11-parameter augmented model (red solid line) and the 27-parameter augmented model (red dotted line).

The wind farm is located at a rather complex terrain site, as shown in Fig. 11. Blue turbines are of type GE1.5sle, black and red turbines are of type GE1.5s, the latter being used as sensing turbines as explained later on. Figure 12 shows a top view of the wind farm, including the turbine identifiers.

Historical 10 min SCADA data was made available for this research for a period of 24 months, throughout the years 2015 and 2016. The recorded turbine yaw orientations exhibit sudden jumps and long term drifts. An ad-hoc algorithm was developed for detecting and correcting these data issues. On average, for each turbine $45\%$ of the data points were missing, while $23\%$ were discarded because of low power output ($< 5$ kW) or rotor speed ($< 1$ rpm). As a result, about $33,700$ data points were available for each turbine. Regarding the missing data points, it is unknown whether the turbines were operating or just not reporting. To avoid eliminating a large fraction of the data set, it was assumed that the turbines were indeed operational and thus shedding wakes. This way, even if recordings of one or more turbines were missing at a specific time instance, the data points of the other turbines could still be used.



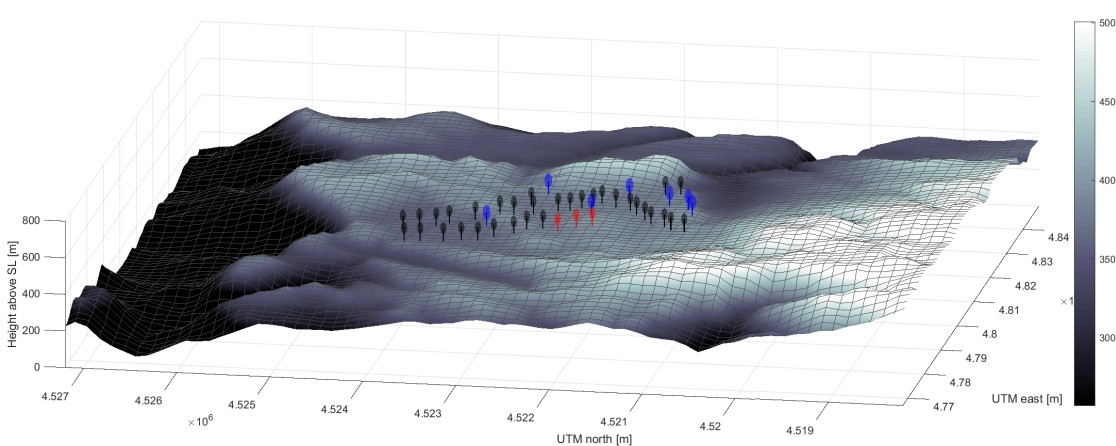

**Figure 11.** 3D view of the Sedini wind farm with terrain elevation, as seen from $\Gamma = 260°$.

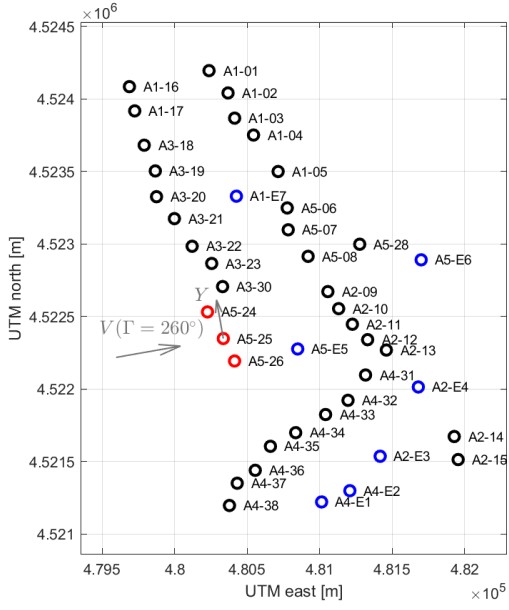

**Figure 12.** Top view of the Sedini wind farm with turbine identifiers. The gray arrows indicate the $X$ and $Y$ axes for an ambient wind direction $\Gamma = 260°$.





As no direct measurements of ambient conditions were available, the method described by Schreiber et al. (2018) was used to identify ambient wind speed and direction. The procedure works as follows. First, the ambient wind direction is estimated from turbine yaw orientations. Second, the ambient wind speed is estimated from the rotor effective wind speed of the free-stream turbines, computed from the turbine power curve below rated wind speed. To this purpose, the three sensing turbines A5-24, A5-25 and A5-26 indicated in red in Fig. 12 were used, checking that they were unwaked by using the flow model; the average of these speeds was attributed to the location of turbine A5-25. This way, $5,667$ ambient wind conditions could be processed for a range of wind directions $\Gamma \in [184°, 320°]$. Based on the ambient wind conditions, the data of all turbines was aggregated in two-dimensional bins: ambient wind speed (bin width of 2 m/s) and ambient wind direction (bin width of $5°$). Figure 13 shows the scaled number of measurements in each bin between 6 and 12 m/s.

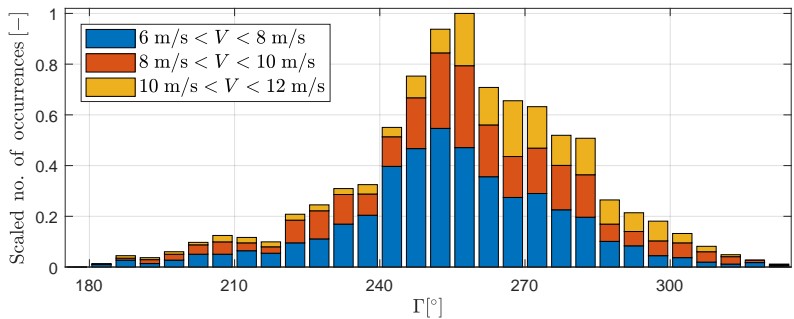

**Figure 13.** Scaled number of measurement data points (10 min mean) within each speed and direction bin.

### 3.2.2 Model setup

Here again the FLORIS implementation was based on the version available online (Doekemeijer and Storm, 2019). The initial values of both the wake and turbulence model parameters were set according to Bastankhah and Porté-Agel (2016) for $(\alpha^*, \beta^*)$, Crespo and Hernández (1996) for $(TI_a^*, TI_b^*, TI_c^*, TI_d^*)$, Niayifar and Porté-Agel (2015) for $(k_a^*, k_b^*)$, and Gebraad et al. (2014) for $(a_d^*, b_d^*)$, as reported in Table 6.

**Table 6.** Initial FLORIS parameters for the Sedini wind farm.

| $\alpha^*$ | $\beta^*$ | $k_a^*$ | $k_b^*$ | $a_d^*$ | $b_d^*$ | $TI_a^*$ | $TI_b^*$ | $TI_c^*$ | $TI_d^*$ |
|------------|-----------|---------|---------|---------|---------|----------|----------|----------|----------|
| 2.32 | 0.154 | 0.3837 | 0.0037 | $-0.0356$ | $-0.01$ | 0.73 | 0.8325 | 0.0325 | $-0.32$ |

The required turbine power and thrust versus wind speed curves were provided by the turbine manufacturer. The vertical shear exponent of the inflow was set to $\alpha_{vs} = 0.143$ and the turbulence intensity to $14\%$, which represent annual average values measured at 65 m of height by an on-site met-mast. Air density was set to the constant value $\rho = 1.177$ kg/m$^3$.

The different turbine foundation heights were accounted for by accordingly increasing the tower heights, using the lowest foundation height as reference (turbine A1-02). Indeed, power measurements of the upstream turbines show a correlation





with the actual turbine hub height with respect to sea level (SL), as shown in Fig. 14. As indicated by the only approximate correlation shown by the figure, it is clear that such simple correction might not provide satisfactory results for all wind directions and all turbines, because complex orthographic flow effects might also play a role. Nonetheless, this approximate correction seems to be a step in the right direction. In addition, some of these effects may be corrected by the lateral non-

uniformity terms added to the augmented model. The reference height of the sheared inflow $z_\mathrm{h}$ (see Eq. (2)) was set to the hub height of the sensing turbine A5-25.

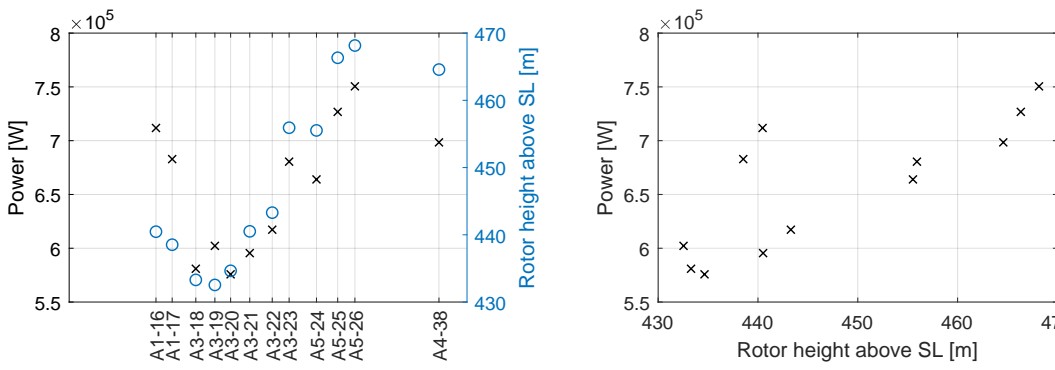

**Figure 14.** Correlation between power output and hub height with respect to SL. Left subplot: power ($\times$ symbols and left $y$ axis) and rotor height above SL ($\circ$ symbols and right $y$ axis) vs. lateral turbine position for a wind direction $\Gamma = 240°$. Right subplot: power vs. rotor height above SL for $\Gamma \in [220°, 275°]$ and $V_\infty \in [8, 10]$ m/s. All conditions are free-stream and all turbines of type GE1.5s.

### 3.2.3 Ranking of correction terms

As for the wind tunnel experiments, here again a first analysis was aimed at ranking the various correction terms. However, since the turbines were operated with a conventional wind-aligned strategy, secondary steering corrections were neglected. The

ranking is based on data points in the range $V \in [6, 10]$ m/s, as described in greater detail in the following subsection.

Figure 15 shows the relative increase of the cost function after optimization eliminating one set of parameters at a time. The results clearly indicate that the non-uniform wind farm inflow speed $\boldsymbol{p}_\mathrm{speed}$ is the most important correction. In fact, this was to be expected, given that the Sedini wind farm is located in a rather complex terrain site. Results indicate also a non-negligible effect of the wake deflection parameters for non-misaligned operation ($a_\mathrm{d}, b_\mathrm{d}$).

On the other hand, the additional model augmentation parameters ($\boldsymbol{p}_\mathrm{TI}, \boldsymbol{p}_\mathrm{winddir}, \boldsymbol{p}_\mathrm{acc}, \boldsymbol{p}_\mathrm{shear}$) do not seem to contribute to a significant extent. Note also the slight retuning of parameters ($\alpha, \beta, k_\mathrm{a}, k_\mathrm{b}$) and ($TI_\mathrm{a}, TI_\mathrm{b}, TI_\mathrm{c}, TI_\mathrm{d}$), which can be explained with the fact that their initial values were taken from the literature, and therefore apply to different turbine types and sites.

Given these results, the rest of the analysis is based only on the sub-set of parameters $\boldsymbol{p}_\mathrm{inflow}$, ($p_{a_\mathrm{d}}, p_{b_\mathrm{d}}$), ($p_\alpha, p_\beta$), ($p_{k_\mathrm{a}}, p_{k_\mathrm{b}}$), ($p_{TI_\mathrm{a}}, p_{TI_\mathrm{b}}, p_{TI_\mathrm{c}}, p_{TI_\mathrm{d}}$). The augmentation term for non-uniform inflow speed is modeled using five nodes along the lateral

position $Y$ located at $[-2000; -1000; 0; 1000; 2000]$ m and six nodes in wind direction $\Gamma$ at $[180; 210; 140; 270; 300; 330]°$,




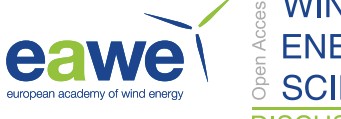

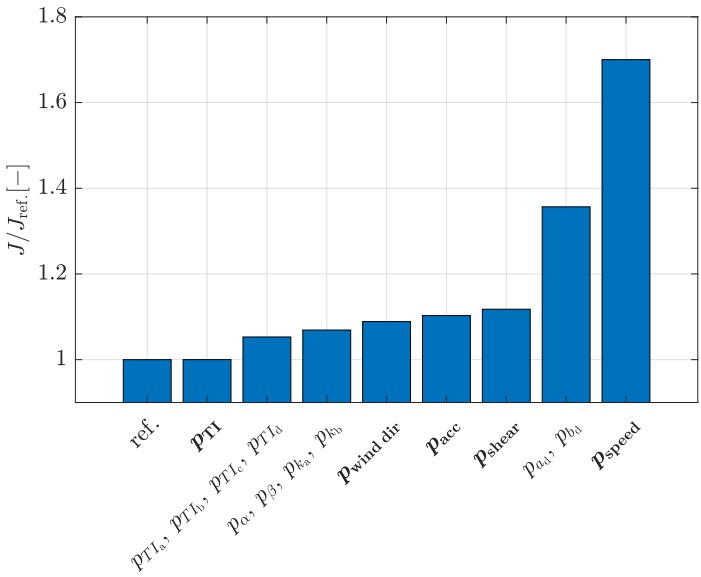

**Figure 15.** Relative increase of the optimization cost function for the Sedini wind farm when eliminating one parameter set at a time.

resulting in 30 nodes. The $Y$-coordinate axis is orthogonal to the wind direction and its origin $Y = 0$ m is located at the position of wind turbine A5-25, as shown in Fig. 12.

The correction parameter definitions, together with their bounds and converged values, are reported in Table 7. Note that all parameters were set to zero at the beginning of the identification process.

**Table 7.** Definition of the parameters, together with their lower and upper bounds, and initial and identified values.

| $i$ | $p_i$ | $p_{\text{lb},i}$ | $p_{\text{ub},i}$ | $p_{\text{init},i}$ | $p_{\text{opt},i}$ | Implementation |
|---|---|---|---|---|---|---|
| $1-30$ | $\boldsymbol{p}_{\text{inflow}}$ | $\mathbf{-0.1}$ | $\mathbf{0.1}$ | $\mathbf{0}$ | see Fig. 16 | $f_{\text{augm,speed}}(Y,Z,\Gamma,\boldsymbol{c}_{\text{speed}},\boldsymbol{p}_{\text{speed}})$ |
| $31$ | $p_\alpha$ | $-\alpha^*$ | $4$ | $0$ | $0.7837$ | $\alpha = \alpha^* + p_\alpha$ |
| $32$ | $p_\beta$ | $-\beta^*$ | $2$ | $0$ | $1.063$ | $\beta = \beta^* + p_\beta$ |
| $33$ | $p_{k_{\text{a}}}$ | $-k_{\text{a}}^*$ | $1$ | $0$ | $-0.2440$ | $k_{\text{a}} = k_{\text{a}}^* + p_{k_{\text{a}}}$ |
| $34$ | $p_{k_{\text{b}}}$ | $-k_{\text{b}}^*$ | $0.1$ | $0$ | $0.01862$ | $k_{\text{b}} = k_{\text{b}}^* + p_{k_{\text{b}}}$ |
| $35$ | $p_{a_{\text{d}}}$ | $-0.5$ | $0.5$ | $0$ | $-0.3169$ | $a_{\text{d}} = a_{\text{d}}^* + p_{a_{\text{d}}}$ |
| $36$ | $p_{b_{\text{d}}}$ | $-0.1$ | $0.1$ | $0$ | $-0.02246$ | $b_{\text{d}} = b_{\text{d}}^* + p_{b_{\text{d}}}$ |
| $37$ | $p_{TI_{\text{a}}}$ | $-TI_{\text{a}}^*$ | $1$ | $0$ | $-0.09577$ | $TI_{\text{a}} = TI_{\text{a}}^* + p_{TI_{\text{a}}}$ |
| $38$ | $p_{TI_{\text{b}}}$ | $-1$ | $1$ | $0$ | $0.3403$ | $TI_{\text{b}} = TI_{\text{b}}^* + p_{TI_{\text{b}}}$ |
| $39$ | $p_{TI_{\text{c}}}$ | $-1$ | $1$ | $0$ | $0.4452$ | $TI_{\text{c}} = TI_{\text{c}}^* + p_{TI_{\text{c}}}$ |
| $40$ | $p_{TI_{\text{d}}}$ | $-1$ | $1$ | $0$ | $-0.3337$ | $TI_{\text{d}} = TI_{\text{d}}^* + p_{TI_{\text{d}}}$ |





### 3.2.4 Results

To identify the 40 parameters of Table 7, only aggregated mean power measurements for wind speeds $V \in [6, 10]$ m/s were used. In addition, only one third of all wind direction bins were employed, with bin centers at $[192.5 : 15 : 312.5]°$, resulting in a total of 9 measurement corridors. The remaining wind direction and speed bins were reserved for validation.

The model outputs $\boldsymbol{y}$ (cf. Eq. (A1)) were defined as

$$
\boldsymbol{y} = \frac{1}{P_{\mathrm{ref}}} \begin{bmatrix} P_{\mathrm{WT1}} \\ ... \\ P_{\mathrm{WT43}} \end{bmatrix},
\tag{11}
$$

where $P_{\mathrm{WT}i}$ is the power of wind turbine $i$ and $P_{\mathrm{ref}} = 1.11$ MW a reference wind turbine value used as scaling factor. A diagonal measurement noise covariance matrix $\boldsymbol{R}$ was used, with all diagonal terms equal to $\sigma^2 = 0.01^2$. The threshold of the highest acceptable variance in the orthogonal parameter estimate was set to $\sigma_t^2 = 0.01$, which corresponds to a relative variance

of $2\%$. The relative weight of each observation was set proportional to the number of measurement points within the respective bin. In a first iteration, 29 orthogonal parameters could be identified. In the second and third iteration only 23 and 25 orthogonal parameters fell below the threshold, although results changed only marginally after the first iteration.

     The identified optimal parameter values $p_{\mathrm{opt},i}$ are included in Table 7 and, for the inflow augmentation, are also reported in Fig. 16. The latter shows, according to the colormap, the inflow augmentation function values $f_{\mathrm{augm,speed}}(Y, \Gamma, \boldsymbol{c}_{\mathrm{speed}}, \boldsymbol{p}_{\mathrm{speed}})$

in the left subplot. Each nodal point is indicated by a circle marker. The figure shows that significant variations in the inflow speed have been detected: for example, considering $\Gamma = 270°$, the inflow speed at $Y = +1000$ m (approximately at the location of wind turbines A3-19/20/21) is $3.5\%$ smaller than the one measured at the reference turbines A5-24/25/26. For the same wind direction, the speed at $Y = -1000$ m (approximately located at the wind turbines A4-36/37/38) is $4.8\%$ larger. These variations are expected to be mainly caused by terrain effects. The right subplot of Fig. 16 shows the parameter uncertainty (Cramér-Rao

bounds). The parameter at the nodal point $(Y = -2000$ m; $\Gamma = 330°)$ is completely unobservable, because it lies far outside of the wind farm perimeter (see Fig. 12). Some of the outer nodal points at $Y = \pm 2000$ m do show significantly increased uncertainties. However, the corresponding augmentation parameters (left subplot) are approximatively zero.

     Figure 17 shows the power coefficient of each individual wind turbine, as indicated by the subplot title, as function of wind direction. The power coefficient is computed as $C_{\mathrm{P}} = P/(0.5\rho A V^3)$, where $\rho = 1.177$ kg/m$^3$ is the constant air density, $A =$

$\pi (70.5/2)^2$ m$^2$ a reference rotor area, and $V$ the corresponding estimated ambient wind speed. Blue crosses indicate SCADA data points, the ones used for identification having been encircled. The gray shaded area indicates the standard deviation within the binned measurements. The FLORIS (non-augmented) power estimates are shown by the black dashed lines, whereas the augmented model results are shown using red solid lines.

     Even though the baseline FLORIS power estimates already exhibit a reasonable correlation with the measurements for

many turbines and wind directions, a significant improvement is achieved by the augmented model. Note that for $\Gamma < 210°$ and $\Gamma > 300°$ the number of measurement points within each bin becomes smaller (see Fig. 13), limiting the measurement quality/trustworthiness. More specifically, the augmented model shows improvements in the modeling of the free stream turbine





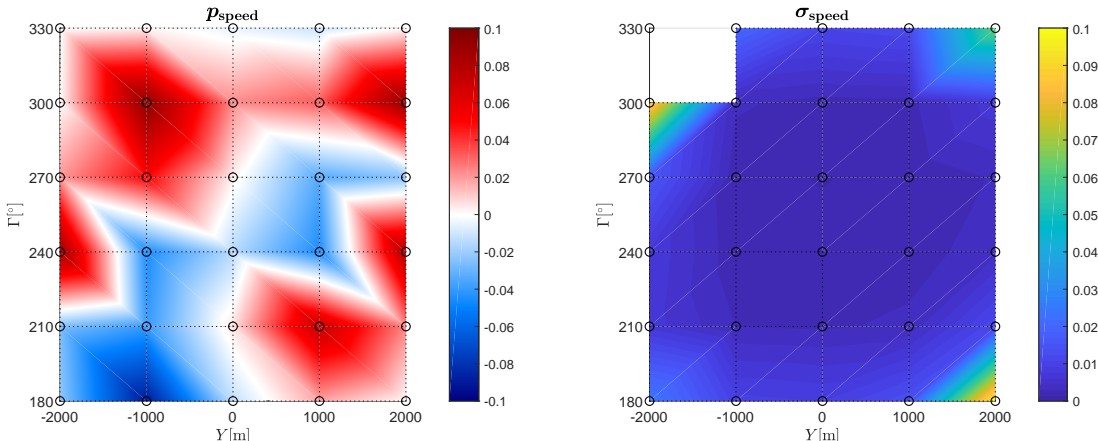

**Figure 16.** Identified inflow augmentation parameters (left subplot) and their uncertainties (right subplot). Nodal points are indicated by the circle markers.

power, due to the effects of the wind farm inflow augmentation terms. Furthermore, the predictions of the wake-induced power deficits are corrected, improving in many cases the deficit depth as well as the deficit location in terms of wind direction.

The same results of Fig. 17 are also presented in a more synthetic form in terms of error probability densities in Fig. 18, where the error is defined as $\epsilon = C_{\mathrm{P,Meas.}} - C_{\mathrm{P,FLORIS/Augm.}}$. Each subplot shows the results for a different wind speed range. Note that the modeling error is reduced also for wind speed ranges that have not been used for model identification (i.e. $V \in [6,8]$ m/s and $V \in [10,12]$ m/s). The overall root mean squared error is reported within the legend, showing error reductions of $14\%$, $22\%$ and $19\%$, respectively, highlighting the generality of the identified model and augmentation parameters.

## 4   Conclusions

This paper has presented a new method to calibrate and augment parametric wind farm models. The proposed approach builds on the vast body of knowledge and experience embedded in available reduced wind farm flow models. However, recognizing that any such model will always have only a limited prediction accuracy, the present approach augments a baseline model with ad-hoc extra terms designed to correct some of its presumed specific deficiencies. These additional elements of the model are then learnt from operational data. Optionally, the baseline model parameters can also be tuned within a single integrated process. By design, the method has been exclusively based here on SCADA power measurements; therefore, it is readily applicable to most operational wind farms, whenever such data is available. However, the concept of model augmentation is very general and could clearly be used also with additional measurements.

To limit the number of free parameters and to overcome the fact that the identification problem can be over-parameterized and hence ill-posed, a parameter transformation into an orthogonal space has been used. Thereby, only parameters that are sufficiently visible within a given data set enter into the identification process.







**Figure 17.** Power coefficient of each individual wind turbine, as indicated by the subplot title, as function of wind direction $\Gamma$ for wind speeds $V \in [8, 10]$ m/s. The gray shaded area indicates the standard deviation within the binned measurements. The number of measurements within each bin is reported in Fig. 13.



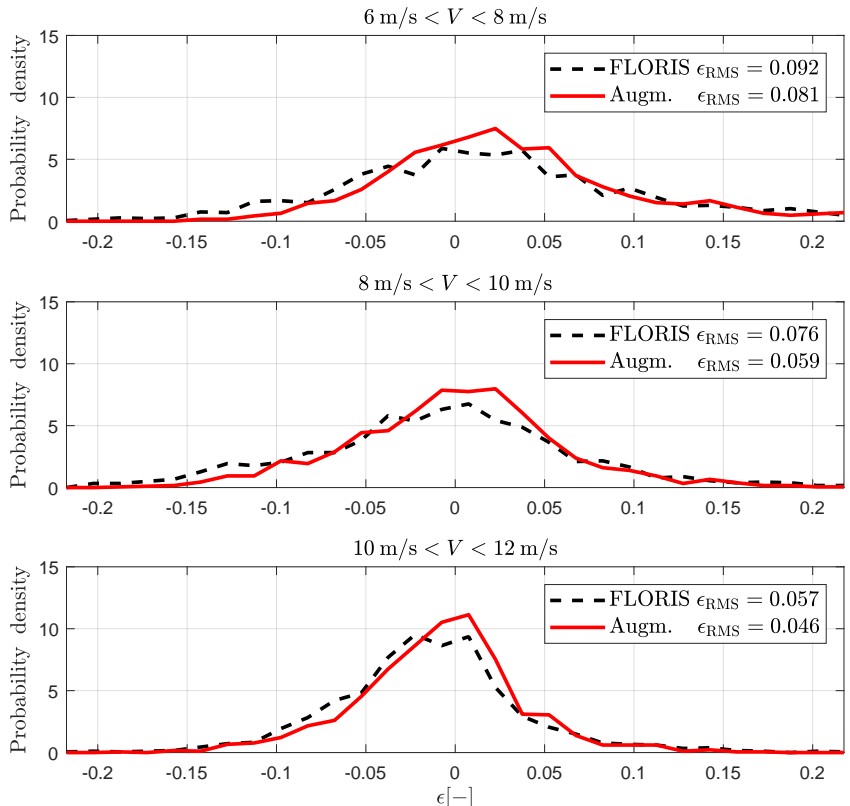

**Figure 18.** Error probability density functions for different wind speed ranges.

The method was first applied to a large data set obtained with scaled wind turbines operating in a boundary layer wind tunnel. Thereby, it was shown that a correct learning of the extra modeling terms is achieved. These conclusions are made possible by the fact that, in this case, the flow and wake characteristics are known with good accuracy. Next, the method was tested on a real wind farm, in a realistic and highly complex situation.

5    Based on the results shown here, the following conclusions can be drawn:

– Within the wind tunnel environment, a correct learning of non-uniform wind farm inflow speed and of secondary steering effects has been achieved. In particular, the latter shows a good match with detailed wake measurements in wind misaligned conditions. It is remarkable, and very promising, that such detailed features of the solution could be inferred purely from power operational data, even when starting from a baseline model that does not consider at all secondary

10    steering.





- The application to field data has shown that, as expected for the complex terrain site analyzed here, orographic effects play a driving role. A marked model improvement could be observed, even in conditions where the model was used for extrapolating outside of the training conditions. It is worth noting that, in many practical onshore applications, orographic effects will be present, and the fact that on can learn them from operational data is very encouraging. Again, it should be

explicitly pointed out that the baseline model did not include any orographic corrections.

- It has been shown that model tuning and the learning of extra correction terms can be achieved simultaneously. This reduces the risk of adapting the baseline parameters beyond their reasonable limits, driven by umodeled physics.

- Although the augmented models show a much improved accuracy with respect to the baseline, some model mismatch still remains. Although these remaining errors may often be caused by issues in the data rather than in the model, additional

improvements are thought to be possible.

Future work will apply the proposed method to other wind farms, to increase confidence in the obtained results. From longer and richer data sets, possibly in conjunction with meteorological reanalyses, it is presumed that yearly and seasonal variations could be observed. The integration of CFD analyses can be used to support and confirm the identification of orographic effects. Attention should also be paid to improved and additional forms of model corrections, including wake overlap models. Finally,

it is worth pointing out again that an improved knowledge of the flow within a wind farm finds applicability in a potentially large range of digitally-driven applications, including wind farm control, lifetime estimation, power forecasting, predictive maintenance and others. Therefore, it is expected that methods of deriving high-accuracy flow predictions in wind farms will be the subject of significant future research efforts.

*Code and data availability.*   A MATLAB implementation of the wind farm model can be obtained by contacting the authors.

**Appendix A: Identification method**

**A1   Maximum Likelihood estimation of model parameters**

A steady-state wind farm model can be mathematically expressed as

$$\boldsymbol{y} = \boldsymbol{f}(\boldsymbol{p}, \boldsymbol{u}), \tag{A1}$$

where $\boldsymbol{f}(\cdot, \cdot, \cdot)$ is the non-linear static function describing the wind farm model, which depends on free parameters $\boldsymbol{p} \in \mathbb{R}^n$.

These parameters can include both wake model parameters and/or model augmentation parameters. The model inputs $\boldsymbol{u} \in \mathbb{R}^{n_\mathrm{u}}$ can include ambient wind conditions (i.e. ambient wind speed, direction, air density, turbulence intensity, etc.) and control inputs (i.e. yaw misalignment, partialization factor, blade pitch, rotor speed, etc. of each turbine). The model outputs $\boldsymbol{y} \in \mathbb{R}^m$ represent quantities of interest for which measurements are available, in the present work typically the power output of each



wind turbine. Experimental observations $z$ of the simulated outputs will in general result in a residual $r \in \mathbb{R}^m$, caused by measurement and process noise (e.g. plant-model mismatch), so that

$$z = y + r. \tag{A2}$$

Note that within this classical formulation, inputs are assumed to be exactly known. A generalized formulation that assumes
also uncertain inputs can be obtained by promoting the inputs to outputs and introducing new state variables (Wang et al., 2020).

Given a set $S = \{z_1, z_2, ..., z_N\}$ of $N$ independent observations, the likelihood function (Jategaonkar, 2015) can be defined as

$$\mathcal{L}(S\big|_{\boldsymbol{p}}) = \prod_{i=1}^{N} p(z_i\big|_{\boldsymbol{p}}), \tag{A3}$$

where $p(\cdot)$ is the probability of $S$ given $\boldsymbol{p}$. Assuming the residuals $r$ with covariance $\boldsymbol{R}$ to be statistically independent within the set of measurements (i.e., $E[r_i r_j{}^T] = \boldsymbol{R}\delta_{i,j}$, where $\delta_{i,j}$ is the Kronecker delta), the likelihood function can be written as (Jategaonkar, 2015)

$$\mathcal{L}(S\big|_{\boldsymbol{p}}) = \left((2\pi)^m \det(\boldsymbol{R}^{-1})\right)^{-N/2} \exp\left(-\frac{1}{2}\sum_{i=1}^{N} r_i{}^T \boldsymbol{R} r_i\right). \tag{A4}$$

Maximizing $\mathcal{L}$ (or minimizing its negative logarithm), a maximum likelihood estimate of the parameters can be obtained as

$$\boldsymbol{p}_{\mathrm{MLE}} = \arg\min_{\boldsymbol{p}} J(\boldsymbol{p}), \tag{A5}$$

where $J(\boldsymbol{p}) = -\ln(\mathcal{L}(S\big|_{\boldsymbol{p}}))$. The measurement noise covariance matrix $\boldsymbol{R}$ can be estimated under mild hypotheses as $\boldsymbol{R} = \sum_{i=1}^{N} r_i{}^T r_i$, yielding $J(\boldsymbol{p}) = \det(\boldsymbol{R})$, leading to an iteration between a solution at given covariance and a covariance update step (Jategaonkar, 2015). However, in this paper the measurement noise covariance matrix is estimated a priori and therefore assumed to be known. The cost function becomes therefore

$$J(\boldsymbol{p}) = \frac{1}{2}\sum_{i=1}^{N} r_i{}^T \boldsymbol{R}^{-1} r_i. \tag{A6}$$

To ensure reasonable and physically viable solutions, parameters can be forced to stay within predefined upper (subscript ub) and lower (subscript lb) bounds, by adding the corresponding inequality constraints $\boldsymbol{p}_{\mathrm{lb}} \leq \boldsymbol{p} \leq \boldsymbol{p}_{\mathrm{ub}}$ to problem (A5). As the parameter values and constraints can differ in magnitude, it is a good practice to scale all parameters such that a value of 1 corresponds to the upper bound $p_{\mathrm{ub}}$ and a value of $-1$ to the lower one $p_{\mathrm{lb}}$. The optimization problem can finally be solved
numerically by a suitable algorithm, such as Sequential Quadratic Programming (SQP) (Nocedal and Wright, 2006).

### A1.1 Identifiability of parameters

The Fisher information matrix $\boldsymbol{F} \in \mathbb{R}^{n \times n}$ is defined as

$$\boldsymbol{F} = \sum_{i=1}^{N} \left[\frac{\partial \boldsymbol{y}_i}{\partial \boldsymbol{p}}\right]^T \boldsymbol{R}^{-1} \left[\frac{\partial \boldsymbol{y}_i}{\partial \boldsymbol{p}}\right], \tag{A7}$$





and describes the curvature of the likelihood function. It can be shown (Jategaonkar, 2015) that a lower bound (termed Cramér-Rao bound) of the covariance of the estimated parameter is given by

$$\boldsymbol{F}^{-1} = \boldsymbol{P} \leq \mathrm{Var}(\boldsymbol{p}_{\mathrm{MLE}} - \boldsymbol{p}_{\mathrm{true}}), \tag{A8}$$

where $\boldsymbol{p}_{\mathrm{true}}$ are the true but unknown parameters. The $k$-th diagonal element of $\boldsymbol{P}$ is a lower bound on the variance of the
$k$-th estimated parameter, while the correlation between different parameters is captured by the off-diagonal terms of that same matrix. The correlation coefficient between two parameters $i$ and $j$ is defined as

$$\Psi_{p_i, p_j} = \frac{P_{i,j}}{\sqrt{P_{i,i} P_{j,j}}}, \tag{A9}$$

where $P_{i,j}$ denotes the $i, j$-th element (row, column) of $\boldsymbol{P}$. By analyzing the estimated parameter variance, as well as the correlation between the parameters, valuable insight into the well-posedness of the parameter identification problem can be
readily obtained.

### A1.2   Problem transformation using the SVD

When some parameters are highly correlated or have large variance, the problem is ill-posed: it might exhibit sluggish convergence, or not converge at all, and small changes in the inputs may lead to large changes in the estimates. Such situations are difficult to solve in the physical space, because parameters are typically coupled together to some degree through the model.

To untangle the parameters, one may resort to the SVD (Golub and van Loan, 2013). By this approach (Hansen, 1987; Waiboer, 2007; Bottasso et al., 2014a), the original parameters are mapped into a new set of uncorrelated (orthogonal) parameters. Since the new unknowns are uncorrelated, one can set a threshold to their variance by using the Cramér-Rao bound, and only retain in the optimization those that are observable within the given data set.

The Fisher matrix $\boldsymbol{F}$ is first factorized as $\boldsymbol{F} = \boldsymbol{M}^T \boldsymbol{M}$, where $\boldsymbol{M} \in \mathbb{R}^{Nm \times n}$ is defined as

$$\boldsymbol{M} = \begin{bmatrix} \boldsymbol{R}^{-1/2} \frac{\partial \boldsymbol{y}_1}{\partial \boldsymbol{p}} \\ \boldsymbol{R}^{-1/2} \frac{\partial \boldsymbol{y}_2}{\partial \boldsymbol{p}} \\ \dots \\ \boldsymbol{R}^{-1/2} \frac{\partial \boldsymbol{y}_N}{\partial \boldsymbol{p}} \end{bmatrix}. \tag{A10}$$

Assuming a larger number of measurements than parameters ($Nm > n$), matrix $\boldsymbol{M}$ can be decomposed into

$$\boldsymbol{M} = \boldsymbol{U} \boldsymbol{\Sigma} \boldsymbol{V}^T, \tag{A11}$$

where $\boldsymbol{U} \in \mathbb{R}^{Nm \times Nm}$ and $\boldsymbol{V} \in \mathbb{R}^{n \times n}$ are the matrices of left and right, respectively, singular vectors, while

$$\boldsymbol{\Sigma} = \begin{bmatrix} \boldsymbol{S} \\ \boldsymbol{0} \end{bmatrix}, \tag{A12}$$

where $\boldsymbol{S} \in \mathbb{R}^{n \times n}$ is a diagonal matrix, whose entries $s_i$ are the singular values sorted in descending order.





By using Eq. (A11) and the factorization of $\boldsymbol{F}$, the inverse of the Fisher information matrix can be written as

$$\boldsymbol{P} = \boldsymbol{V}\boldsymbol{S}^{-2}\boldsymbol{V}^{T}. \tag{A13}$$

Note that the columns of the orthogonal matrix $\boldsymbol{V}$ are also the eigenvectors of $\boldsymbol{P}$ and $s_i^{-2}$ the corresponding eigenvalues. Furthermore, $\boldsymbol{P}$ and $\boldsymbol{F}$ are symmetric and, based on the spectral theorem, diagonalizable.

The physical parameters $\boldsymbol{p}$ can now be transformed into a new set of orthogonal parameters $\boldsymbol{\Theta}$ by a rotation performed with the right singular values:

$$\boldsymbol{\Theta} = \boldsymbol{V}^{T}\boldsymbol{p}. \tag{A14}$$

For the transformed set of parameters, the Cramér-Rao bound on the variance of the estimates is the diagonal matrix $\boldsymbol{S}^{-2} \leq$ Var$(\boldsymbol{\Theta}_{\mathrm{MLE}} - \boldsymbol{\Theta}_{\mathrm{true}})$. Therefore, a small singular value $s_i$ corresponds to a large uncertainty in the corresponding orthogonal

parameter estimation.

To remove parameters that cannot be estimated with sufficient accuracy, matrix $\boldsymbol{S}$ can be partitioned as

$$\boldsymbol{S} = \begin{bmatrix} \boldsymbol{S}_{\mathrm{ID}} & \boldsymbol{0} \\ \boldsymbol{0} & \boldsymbol{S}_{\mathrm{NID}} \end{bmatrix}, \tag{A15}$$

where $\boldsymbol{S}_{\mathrm{ID}}$ contains the identifiable singular values, i.e. those such that $s_i^{-2} < \sigma_t^2$, $\sigma_t$ being a threshold on the highest acceptable standard deviation in the estimate. On the other hand, matrix $\boldsymbol{S}_{\mathrm{NID}}$ contains singular values associated with parameters that can-

not be identified with sufficient accuracy, and are therefore discarded. Accordingly, $\boldsymbol{V}$ is also partitioned as $\boldsymbol{V} = [\boldsymbol{V}_{\mathrm{ID}}, \boldsymbol{V}_{\mathrm{NID}}]$, while the orthogonal parameters are partitioned as $\boldsymbol{\Theta} = [\boldsymbol{\Theta}_{\mathrm{ID}}^{T}, \boldsymbol{\Theta}_{\mathrm{NID}}^{T}]^{T}$. Finally, the physical parameters are expressed in terms of the sole identifiable orthogonal parameters:

$$\boldsymbol{p} \approx \boldsymbol{V}_{\mathrm{ID}}\boldsymbol{\Theta}_{\mathrm{ID}}. \tag{A16}$$

Given that the Fisher matrix depends on the values of the parameters $\boldsymbol{p}$, an iterative procedure should be followed, where the

diagonalization of the problem is repeated at each update of the parameter vector.

### A1.3   Identification method with variable measurement weights

In some cases, it may be useful to increase the importance of some measurements in the parameter estimation problem. This can be readily obtained by simply treating an observation with weight $w$ as if it appeared $w$ times in the observation data set (Karampatziakis and Langford, 2011). Cost function (A6) then becomes

$$J(\boldsymbol{p}) = \frac{1}{2}\sum_{i=1}^{N} w_i {\boldsymbol{r_i}}^{T} \boldsymbol{R}^{-1}\boldsymbol{r_i}, \tag{A17}$$

where $w_i$ is the relative weight of observation $i$ and $\sum_{i=1}^{N} w_i = N$. Similarly, the Fisher matrix becomes

$$\boldsymbol{F} = \sum_{i=1}^{N} w_i \left[\frac{\partial \boldsymbol{y_i}}{\partial \boldsymbol{p}}\right]^{T} \boldsymbol{R}^{-1}\left[\frac{\partial \boldsymbol{y_i}}{\partial \boldsymbol{p}}\right], \tag{A18}$$





and its factorization is

$$
M = \begin{bmatrix} \sqrt{w_1} R^{-1/2} \frac{\partial y_1}{\partial p} \\ \sqrt{w_2} R^{-1/2} \frac{\partial y_2}{\partial p} \\ ... \\ \sqrt{w_N} R^{-1/2} \frac{\partial y_N}{\partial p} \end{bmatrix}. \tag{A19}
$$

The remainder of the formulation is not affected by the introduction of weights.

## Appendix B: Extended wind tunnel results

Figures B1, B2 and B3 report the power outputs of WT1, WT2 and WT3, respectively, for all tested configurations. In each figure, clusters of three subplots represent a unique turntable position, as indicated by the title and the wind farm layout sketch therein. The left part of each subplot shows the turbine power coefficient $C_{P,WTi}$ as a function of $\gamma_{WT1}$ ($x$-axis) and $\gamma_{WT2}$ ($y$-axis). All measured configurations are indicated by a small cross symbol, whereas the measurements used for parameter identification are circled. The central part of each subplot shows the FLORIS model error $\epsilon_{FLORIS} = C_{P,Meas.} - C_{P,FLORIS}$,

including an annotation of the root mean squared error $\epsilon_{RMS}$. Similarly, the right part of each subplot shows the augmented model error $\epsilon_{Augm.}$.

For the first upstream wind turbine, WT1, the baseline FLORIS shows significant errors depending on the turntable position. For $\gamma_{TT} < 0°$ the model under-predicts turbine power because of the lack of uniformity of the flow, as also shown in Fig. 7. The opposite behaviour can be seen for $\gamma_{TT} > 0°$. The augmented model however shows significant improvements, which are due

to the inflow correction. Still some under-prediction for $\gamma_{TT} = -11.5°$ is present, which is probably caused by an excessively small number of parameters in the inflow augmentation function and/or by the third wind turbine power measurements, which are also strongly affected by lateral inflow variations.

The power of WT2, shown in Fig. B2, is only weakly affected or improved by the model corrections. In fact, in all investigated conditions, the second turbine lateral position remains almost constant, such that the inflow correction does not have

much direct effect. However, secondary steering only slightly changes the inflow direction at WT2; for example, as shown in Fig. 8, a $20°$ misalignment of WT1 changes the wind direction by about $1.9°$. This leads to small misalignments and thus only very small changes in power output considering the cosine-law. In addition, secondary steering leads also to a slight lateral deflection of the non-uniform inflow.

The power of WT3, reported in Fig. B3, shows significant improvements when using the augmentation terms. For example,

for $\gamma_{TT} > 0°$ the baseline model under-predicts the real flow velocities —and hence the power output— at WT3, an error that is corrected by the augmented model. In addition, for $| \gamma_{WT1} | > 0$, secondary steering augmentation affects the deflection of the second turbine wake (as shown in Fig. 8), leading to further improvements.



**Figure B1.** Wind turbine WT1. Each cluster of three subplots represents a unique turntable position, as indicated by the title and the wind farm layout sketch. Left subplot: turbine power coefficient $C_{P,WT1}$ as a function of $\gamma_{WT1}$ ($x$-axis) and $\gamma_{WT2}$ ($y$-axis). Middle subplot: FLORIS model error. Right subplot: augmented model error. Cross symbols: all measured configurations. Circles: conditions used for parameter identification.



**Figure B2.** Wind turbine WT2. Each cluster of three subplots represents a unique turntable position, as indicated by the title and the wind farm layout sketch. Left subplot: turbine power coefficient $C_{P,WT2}$ as a function of $\gamma_{WT1}$ ($x$-axis) and $\gamma_{WT2}$ ($y$-axis). Middle subplot: FLORIS model error. Right subplot: augmented model error. Cross symbols: all measured configurations. Circles: conditions used for parameter identification.



**Figure B3.** Wind turbine WT3. Each cluster of three subplots represents a unique turntable position, as indicated by the title and the wind farm layout sketch. Left subplot: turbine power coefficient $C_{P,WT3}$ as a function of $\gamma_{WT1}$ ($x$-axis) and $\gamma_{WT2}$ ($y$-axis). Middle subplot: FLORIS model error. Right subplot: augmented model error. Cross symbols: all measured configurations. Circles: conditions used for parameter identification.



*Author contributions.*  JS conducted the main research work. CLB developed the core idea of model augmentation, the solution methodology to implement it and supervised the whole research. JS and CLB wrote the manuscript. BS pre-processed the field measurements. FC was responsible for the execution of the wind tunnel tests. All authors provided important input to this research work through discussions, feedback and by improving the manuscript.

5 *Competing interests.*  The authors declare that they have no conflict of interest.

*Acknowledgements.*  The authors express their gratitude to Enel Green Power S.p.A., which granted access to the field data, and to Stefan Kern of GE Renewable Energy, who helped with the data post-processing. The scaled experiments were conducted with the technical support of Alessandro Croce and the staff of the wind tunnel at Politecnico di Milano, and of Robin Weber of TUM; their help is gratefully acknowledged. This work has been supported by the CL-WINDCON project, which receives funding from the European Union Horizon 10  2020 research and innovation program under grant agreement No. 727477, and by the CompactWind II project (FKZ: 0325492G), which receives funding from the German Federal Ministry for Economic Affairs and Energy (BMWi).





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
