# Peer review of "Improving wind farm flow models by learning from operational data"

_Wind Energy Science, 2019_

## Referee Comment (RC1) · Anonymous Referee #1 · 6 Jan 2020

Thank you for this submission. I found the concepts presented in this paper to be very interesting, and very convincingly presented. I strongly agree with the authors proposal, laid out in the introduction, that a compromise of using engineering models, with corrections learned from SCADA data is a good way to maintain the advantages of model-based and data-driven approaches. The authors make a convincing case, using wind tunnel and SCADA studies that the approach outlined in the paper successfully delivers these benefits. Therefore, I believe the paper is of high practical value as wake control, based on engineering models, is increasingly deployed in field campaigns.

An additional overall comment is that the introduction provides good motivations behind developing these techniques, and the references to existing literature and putting this work into context is very well done.

General Comments:

I wasn't clear on the concept of node locations (for example c(.) p(.)) in equation 3). Do these node correspond to specific locations, for example turbine locations? If they are defined wrt inflow wind direction, do they need to rotate with wind direction?

Another point of confusion related to which online versions of tools is being discussed. FLORIS itself is available but the citation for FLORIS is a paper from Doekemeijer. Is this a specific version of the code, and also, is it also available online? Further, are the tools, modifications discussed in this paper available anywhere online as well?

Could you speak a little bit to one question I wondered about? If each parameter is assigned a normal tuning parameter, as well as a correction term, is there a not an issue of non-uniqueness, and an infinite (maybe bounded by the tuning parameter) set of identical options? Is there a danger also of over-fitting given the expanded set of parameters?

Specific Comments:

Page 6, line 25: "is the lateral distance to the wake centerline", isn't this made complex in wake steering if we assume effects such as curl, to define the wake centerline? Or is there an "effective" centerline (such as position of minimum production?)

Fig 2: Would it be possible to note the node locations in this figure?

Page 15: Ref to Wang paper, will this paper be available soon?

Fig 7: Is there any meaning to the x's being on the f = -0.2 line or is this just an obstructed place?

Conclusion: I understand this method mostly from fitting to a single direction and learning the corrections, but does it still afford a wind rose type calculation across wind speeds and directions?

---

## Referee Comment (RC2) · Anonymous Referee #2 · 14 Feb 2020

In this manuscript, the authors propose a method to improve the calibration of wind farm models, specifically FLORIS in this work, by using power measurements of the wind turbines. After the introduction providing motivations for this work, the methodology is quickly summarized in Sect. 2. The method is then applied for two cases, namely a wind tunnel test with three turbine models and a real onshore wind farm on complex terrain.

I believe this work is highly relevant for the wind energy community and it presents some novel and interesting results. The "augmentation" of the model parameters is not a trivial problem and here is well presented and these preliminary results are convincing. My main criticism is about the organization of the paper. I believe that the methodology for the model calibration, which is currently described in Appendix A,

should be moved to the main text body and merged with Sect. 2. I personally jumped from Sect. 2 to Appendix A, before reading Sect. 3, otherwise it was difficult for me to fully understand, for instance, the parameters provided in Table 2, the observability of the parameters, why the parameters suddenly become "orthogonal parameters "at L12 of page 14, what is the transformation matrix reported in Table 3 or the correlation coefficient matrix of Table 4. Therefore, my suggestion, in general, is to enhance the readability of this manuscript even for readers who are not experts on this kind of technique. More detailed comments are reported below.

1. P1L17, "This paper describes a new method to estimate turbine inflow within a wind farm.". This does not seem to be the objective of the work, at least not the main one. 2. P3L7, cross-check this sentence... "can help clarify" ... 3. P3L25, cross-check 4. P4L12, provide more details on how you calculate turbine thrust. 5. P6L28, can you provide a more detailed explanation of why this correction is performed with two Gaussian functions? Then, clarify if you mean sum or difference of these two functions, see Eq. 5. 6. P7L16, cross-check 7. P8L2, provide references for these aisle jets. Is this a new terminology or it has already been used in literature? 8. Table 1, please clarify how these initial parameters are estimated. 9. P13L5, the values of Cspeed should be provided in a non-dimensional form. 10. Table 2, how did you select these bound values? 11. P14L12, in the text is not clear how you switch from the actual model parameters to the orthogonal parameters. It becomes clear only after reading Appendix A. 12. P14L14, Similarly to the concept of observability. 13. P14L18, check on in Fig. 7 14. P15L18, check the new line

---

## Author Comment (AC1) · 31 Mar 2020

**Reply to Reviewers**

We thank the reviewers who, with their detailed analyses and constructive inputs, have improved the quality of this paper. A list of point-by-point replies to their comments is reported in the following, and a revised version of the manuscript is attached with highlighted changes.

In addition, we have taken the opportunity for several minor improvements to the text to increase clarity or form.

Best regards, The authors

**Reviewer 1:**

**Reviewer**: Thank you for this submission. I found the concepts presented in this paper to be very interesting, and very convincingly presented. I strongly agree with the authors proposal, laid out in the introduction, that a compromise of using engineering models, with corrections learned from SCADA data is a good way to maintain the advantages of model-based and data-driven approaches. The authors make a convincing case, using wind tunnel and SCADA studies that the approach outlined in the paper successfully delivers these benefits. Therefore, I believe the paper is of high practical value as wake control, based on engineering models, is increasingly deployed in field campaigns.

An additional overall comment is that the introduction provides good motivations behind developing these techniques, and the references to existing literature and putting this work into context is very well done. **Authors**: Thank you for your words and positive feedback.

General Comments:

**Reviewer:** I wasn't clear on the concept of node locations (for example c(.) p(.)) in equation 3). Do these node correspond to specific locations, for example turbine locations? If they are defined wrt inflow wind direction, do they need to rotate with wind direction?

**Authors:** We improved the text to clarify the definition of the correction terms, including nodes and nodal values.

**Reviewer:** Another point of confusion related to which online versions of tools is being discussed. FLORIS itself is available but the citation for FLORIS is a paper from Doekemeijer. Is this a specific version of the code, and also, is it also available online? Further, are the tools, modifications discussed in this paper available anywhere online as well?

**Authors:** The citation for the FLORIS framework in section 2.1 describes the model equations and therein a reference to the Github repository is given (Doekemeijer, 2018; reference 42). Due to updates in this repository, the work on wind tunnel experiments used a different version of the software (Doekemeijer and Storm, 2018) than the one on field data (Doekemeijer and Storm, 2019). A software implementation of the methods described here is available by contacting the authors, as noted at the end of the paper.

**Reviewer:** Could you speak a little bit to one question I wondered about? If each parameter is assigned a normal tuning parameter, as well as a correction term, is there a not an issue of non-uniqueness, and an infinite (maybe bounded by the tuning parameter) set of identical options? Is there a danger also of overfitting given the expanded set of parameters? **Authors:** If the reviewer is referring to Eq. (1), then there is a misunderstanding: the initial values indicated with  $k^*$  are held fixed, while only the correction parameters  $p_k$  are assumed as unknown and identified. The text was improved to clarify this point. On the other hand, if the reviewer refers to the general concept of model augmentation, then it should be remarked that the original model parameters and the extra correction terms have a different functional form in the augmented governing equations. Hence, they should be distinguishable from each other, as they imply different effects on the model. Having said this, it is however in general impossible to guarantee the uniqueness of the solution. This is indeed the reason why we employ the special SVD-based identification, which is capable of highlighting the collinearity of group of parameters. This problem is addressed first in the introduction and then explained in detail in sections "Identifiability of parameters" and "Problem transformation using the SVD" (now moved from the appendix into Section 2). The introduction has been slightly rephrased to clarify this point.

**Specific Comments:**

**Reviewer:** Page 6, line 25: "is the lateral distance to the wake centerline", isn't this made complex in wake steering if we assume effects such as curl, to define the wake centerline? Or is there an "effective" centerline (such as position of minimum production?)

**Authors:** The baseline wind farm model used in this work has a wake centerline definition, which corresponds to the minimum velocity within the wake. We rephrased the text to clarify this point.

**Reviewer:** *Fig 2: Would it be possible to note the node locations in this figure?* **Authors:** The figure was updated with the inflow node locations.

**Reviewer: Page 15: Ref to Wang paper, will this paper be available soon?**

**Authors:** The paper Wang et al., 2020 is in its final stages of internal revision, and will soon be submitted to Wind Energy Science. However, we eliminated this sentence and the reference, as we cannot guarantee that the paper by Wang will be available before the present one is published.

**Reviewer:** Fig 7: Is there any meaning to the x's being on the f = -0.2 line or is this just an obstructed place? **Authors:** There's no meaning, and the markers have been moved to the lower edge of the figure.

**Reviewer:** Conclusion: I understand this method mostly from fitting to a single direction and learning the corrections, but does it still afford a wind rose type calculation across wind speeds and directions? **Authors:** Indeed yes, inflow corrections depend on wind direction because they account for orographic effects that are clearly different depending on where the wind is blowing from. This aspect has been further clarified in the text, and the whole explanation of the correction term expressed by Eq. (2) has been re-written.

**Reviewer 2:**

**Reviewer:** In this manuscript, the authors propose a method to improve the calibration of wind farm models, specifically FLORIS in this work, by using power measurements of the wind turbines. After the introduction providing motivations for this work, the methodology is quickly summarized in Sect. 2. The method is then applied for two cases, namely a wind tunnel test with three turbine models and a real onshore wind farm on complex terrain.

I believe this work is highly relevant for the wind energy community and it presents some novel and interesting results. The "augmentation" of the model parameters is not a trivial problem and here is well presented and these preliminary results are convincing. My main criticism is about the organization of the

paper. I believe that the methodology for the model calibration, which is currently described in Appendix A, should be moved to the main text body and merged with Sect. 2. I personally jumped from Sect. 2 to Appendix A, before reading Sect. 3, otherwise it was difficult for me to fully understand, for instance, the parameters provided in Table 2, the observability of the parameters, why the parameters suddenly become "orthogonal parameters "at L12 of page 14, what is the transformation matrix reported in Table 3 or the correlation coefficient matrix of Table 4. Therefore, my suggestion, in general, is to enhance the readability of this manuscript even for readers who are not experts on this kind of technique. More detailed comments are reported below.

**Authors:** Thank you for your positive feedback and comments. We appreciate your main criticism about the organization of the paper and followed your proposal to include Appendix A into the main body in Section 2.

**Reviewer:** 1. P1L17, "This paper describes a new method to estimate turbine inflow within a wind farm.". This does not seem to be the objective of the work, at least not the main one.

**Authors:** We rephrased the sentence to "This paper describes a new method to improve a wind farm flow model directly from standard operational data.", which repeats the title and is the objective of this paper.

**Reviewer:** 2. *P3L7, cross-check this sentence... "can help clarify" ...* **Authors:** The sentence was modified.

Reviewer: *3. P3L25, cross-check* Authors: Thank you.

**Reviewer:** *4. P4L12, provide more details on how you calculate turbine thrust.* **Authors:** This is explained in the model setup Sections 3.1.2 (wind tunnel) and 3.2.2 (full-scale). We slightly rephrased the relevant parts.

**Reviewer:** 5. P6L28, can you provide a more detailed explanation of why this correction is performed with two Gaussian functions? Then, clarify if you mean sum or difference of these two functions, see Eq. 5. **Authors:** The explanation included already in the paper has been updated to: "This particular choice of shape functions is motivated by the results shown in Fig. 8b of Wang et al. (2018). Indeed, large-eddy simulations and measurements reveal the presence of a stronger lateral velocity component directed towards the wake on the leeward side of the wake itself, and of an opposite and weaker lateral component on the windward side. Such a distribution can be approximated by two Gaussian functions using Eq. (5). "

Figure 8: Wake of a single yawed wind turbine: (a) stream-wise velocity (b) lateral velocity. Dashed lines indicate position and orientation of the downstream machines in the cluster. Negative values of the lateral velocity indicate a velocity directed downwards in the picture.

Regarding the question on the sum or difference, in Eq. (5) we mean the difference of the two Gaussian functions, and therefore we have changed the text accordingly.

**Reviewer:** *6. P7L16, cross-check* **Authors:** Thank you.

**Reviewer:** 7. *P8L2, provide references for these aisle jets. Is this a new terminology or it has already been used in literature?*

**Authors:** It is an already used term, and aisle jets (whose meaning is explained in our paper) are mentioned in Xie and Archer, 2017, which is one of the sources used here to discuss this topic.

**Reviewer:** 8. Table 1, please clarify how these initial parameters are estimated.

**Authors:** All baseline model parameters, reported in Table 1, are taken from Campagnolo et al. (Campagnolo, 2019), where they were identified based on single turbine wake measurements. The text was updated to clarify this point.

**Reviewer:** *9. P13L5, the values of Cspeed should be provided in a non-dimensional form.* **Authors:** We now included the non-dimensional node locations.

**Reviewer:** *10. Table 2, how did you select these bound values?* **Authors:** The initial parameters and bound values have been chosen by an educated guess, as now noted in the text.

**Reviewer:** 11. P14L12, in the text is not clear how you switch from the actual model parameters to the orthogonal parameters. It becomes clear only after reading Appendix A. **Authors:** We reorganized the paper.

**Reviewer:** *12. P14L14, Similarly to the concept of observability.* **Authors:** We reorganized the paper.

**Reviewer:** *13. P14L18, check on in Fig. 7* **Authors:** Thank you.

**Reviewer:** *14. P15L18, check the new line* **Authors:** Thank you.

[revised manuscript text omitted]
  $p_{\text{speed}}$  and bilinear shape functions with node locations espeed and nodal values pspeed. Note that Eq. interpolate the error in each cell based on the nodal values at its corners. Equation (2) could be extended to include also a longitudinal wind-aligned coordinate, similarly to the localized speedup ratios of Jacobsen (2019). For simplicity, the present correction does not include the operating conditions of the downstream machines that, in principle, would be necessary in order to more accurately, to model wind farm blockage effects. Therefore, the present correction can be interpreted as a primarily orography-induced one.

Local orographic effects and blockage may also induce variability of in the wind direction  $\Gamma$ . Similarly, the vertical shear exponent  $\alpha_{\rm vs}$  and turbulence intensity I may vary, for example on account of non-uniform roughness induced by vegetation or other obstacles. To include these effects in the farm flow model, the baseline quantities are augmented as

$$\Gamma(Y) = \Gamma_{\rm ref} + Y f_{\rm augm,dir}(\Gamma_{\rm ref}, \boldsymbol{c}_{\rm dir}, \boldsymbol{p}_{\rm dir}), \tag{3a}$$

$$\alpha_{\rm vs}(\Gamma) = \alpha_{\rm vs,ref} + f_{\rm augm,shear}(\Gamma, \boldsymbol{c}_{\rm shear}, \boldsymbol{p}_{\rm shear}), \tag{3b}$$

. . . .

$$I(\Gamma) = I_{\text{ref}} + f_{\text{augm},I}(\Gamma, \boldsymbol{c}_{\text{I}}, \boldsymbol{p}_{\text{I}}).$$
(3c)

In all these expressions,  $(\cdot)_{ref}$  indicates a baseline reference quantity, while function  $f_{augm,(\cdot)}$  is a correction termbased here on . This function is defined on the 1D space  $\Gamma \in [\Gamma_{\min}, \Gamma_{\max}]$ , discretized with nodes  $c_{(i)} = [\dots; \Gamma_i; \dots]_{(i)}$ , using linear shape functions, with  $e_{(.)}$  to interpolate the corresponding nodal values  $p_{(.)}$ . Here again, by selecting  $\Gamma_{\min}$  and  $p_{(.)}$  the corresponding node locations and nodal values, respectively  $\Gamma_{\max}$  corrections can be applied to the whole wind rose or just to a sector.

**Secondary steering.** By misaligning a wind turbine rotor with respect to the incoming flow direction, the rotor thrust force 25 is tilted, thereby generating a cross-flow force that laterally deflects the wake. As shown with the help of numerical simulations by Fleming et al. (2018), this cross-flow force induces two counter rotating vortices that, combining with the wake swirl induced by the rotor torque, lead to a curled wake shape. As observed experimentally by Wang et al. (2018), the effects of these vortices result in additional lateral flow speed components, which are not limited to the wake itself but extend also outside of it. By this phenomenon, the flow direction within and around a deflected wake is tilted with 30 respect to the upstream undisturbed direction. Therefore, when a turbine is operating within or close to a deflected wake, its own wake undergoes a change of trajectory --termed secondary steering--- induced by the locally modified wind direction. Although models of this phenomenon are being developed (Martínez-Tossas et al., 2019), they significantly

15

5

10

increase the computational cost and are not yet available in standard implementations of engineering wake models as the one used here.

The change of wind direction  $\Delta\Gamma$  at a downstream turbine induced by secondary steering (indicated by the subscript ss) is modeled here as

$$\Delta\Gamma(y) = f_{\text{augm,ss}}(\underline{Y - y_{\text{wc}}}\tilde{y}, \Gamma_{\text{init}}, \boldsymbol{p}_{\text{ss}}), \tag{4}$$

where  $f_{\text{augm,ss}}$  is the correction term and  $\tilde{y} = Y - y_{\text{wc}}$  is the lateral distance to the wake centerline

(see Fig. 1), defined in the baseline wind farm model as the locus of the points of minimum flow speed. According to the notation used in Eq. (6.12) of Bastankhah and Porté-Agel (2016),  $\Gamma_{init}$  indicates the initial wake direction of the closest upstream turbine. The correction term is expressed as the sum difference of two Gaussian functions, and more precisely

$$f_{\text{augm,ss}}(\tilde{y}, \Gamma_{\text{init}}, \boldsymbol{p}_{\text{ss}}) = \Gamma_{\text{init}}\left(p_{\text{ss},1} \exp\left(-0.5\left(\frac{\tilde{y} + \text{sgn}(\Gamma_{\text{init}})p_{\text{ss},3}}{p_{\text{ss},2}}\right)^2\right) - p_{\text{ss},4} \exp\left(-0.5\left(\frac{\tilde{y} + \text{sgn}(\Gamma_{\text{init}})p_{\text{ss},6}}{p_{\text{ss},5}}\right)^2\right)\right), \quad (5)$$

where  $p_{ss} = (p_{ss,1}, p_{ss,2}, p_{ss,3}, p_{ss,4}, p_{ss,5}, p_{ss,6})$  is the vector of free parameters, where parameters 1 and 4 are related to the amplitude, 3 and 6 to the standard deviation, and 2 and 5 to the location of the correction functions. As Since the Gaussian functions are not centered at the wake centerline and the effect of secondary steering is assumed to be symmetric with respect to the misalignment angle, the correction term depends also on the direction of wake deflection  $sgn(\Gamma_{init})$ .

This particular choice of the shape functions is motivated by the experimental results shown in Fig. 8b of Wang et al. (2018). Indeed, LES simulations and measurements reveal the presence of a lateral wake velocity whose maximum is displaced with respect to the wake centerline, as well as a slight lateral flow in the opposite direction that motivates the use of the second Gaussian function in the correction term introduced here.stronger lateral velocity component directed towards the wake on the leeward side of the wake itself, and of an opposite and weaker lateral component on the windward side. Such a distribution can be approximated by two Gaussian functions using Eq. (5).

Note that the change in local wind direction also leads to a slight lateral deflection of the non-uniform wind farm inflow introduced previously. More precisely, for a turbine that is located  $\Delta X$  behind an upstream turbine, the non-uniform inflow expressed by Eq. (2) is evaluated at  $Y + \Delta X \sin(\Delta \Gamma)$  instead of Y.

The upper subplot of Fig. 1 shows the hub height flow speed for two wind turbines modeled in FLORIS, the turbine rotor disks being indicated with thick black lines. The wake centerlines and the undisturbed free stream wind direction are indicated by black dotted and dashed lines, respectively. The upstream turbine is misaligned with respect to the incoming flow and therefore its wake is deflected laterally. Using the baseline wake model, the downstream turbine wake develops along the free stream wind direction. The lower subplot of the same figure shows the

effects of the secondary steering correction term presented above.given by Eq. (5). The plot clearly shows that the downstream turbine wake path is affected by the locally changed wind direction.

10

5

20

15

25

**Figure 1.** Effect of secondary steering on the trajectory of a downstream turbine. Top subplot: baseline wake model; lower subplot: baseline model augmented with the empirical correction term of Eq. (5).

**Non-Gaussian wake and flow acceleration.** Engineering wake models are based, among other hypotheses, on assumed shapes of the speed deficit. For example, the present baseline model assumes a Gaussian distribution of the speed deficit within the wake. Another assumption is that the flow outside the wake is undisturbed, and equal to the free-stream. However, these assumption assumptions can at times not be exactly satisfied, as already observed by Xie and Archer (2017) and Martínez-Tossas et al. (2019), among others. For example, aisle jets are local accelerations of the flow outside of the wake, produced by local blocking in the neighborhood of an operating turbine. It has been reported that aisle jets can induce local flow speedups in excess of 10% of the undisturbed inflow (Dörenkämper et al., 2015).

To account for such effects, the wake velocity  $V_{\rm wake}$  of the baseline model is corrected as

$$V_{\text{wake}}(d_{\text{wc}}) = V_{\text{wake},\text{FLORIS}}(d_{\text{wc}}) \Big( 1 + f_{\text{augm},\text{acc}}(d_{\text{wc}}, \boldsymbol{c}_{\text{acc}}, \boldsymbol{p}_{\text{acc}}) \Big), \tag{6}$$

- 10 where  $V_{\text{wake,FLORIS}}$  is the baseline Gaussian wake speed profile,  $d_{\text{wc}}$  is the absolute distance to the wake center (which, at hub height, is equivalent to  $|\tilde{y}|$ ), and  $f_{\text{augm,acc}}$  represents the correction term, modeled here as which —similarly to the previous corrections— is modeled with linear shape functions characterized by  $e_{\text{acc}}$  node locations and  $p_{\text{acc}}$  nodal values node locations  $e_{\text{acc}}$  (in terms of  $d_{\text{wc}}$ ) and nodal values  $p_{\text{acc}}$ .
- Reduced power extraction due to non-uniform wind turbine inflow. Numerical simulations conducted in FAST (Jonkman and Jonkman, 2018) using its Blade Element Momentum (BEM) implementation yielded a slight reduction in the rotor power coefficient for horizontally sheared flow, when compared to unsheared conditions with the same hub wind speed. Even though BEM can only give a rough indication for such an effect, a correction of the power coefficient of the baseline model is introduced here in the form

$$C_{\rm P} = C_{\rm P,\kappa=0} \left( 1 + p_{\kappa} \kappa^2 \right),\tag{7}$$

where  $C_{P,\kappa=0}$  is the nominal power coefficient,  $\kappa$  the equivalent horizontal linear shear coefficient on the rotor disk and  $p_{\kappa}$  the free correction parameter. The linear shear  $\kappa$  is either due to a lack of lateral uniformity of the inflow or to the impingement of a wake, and it is evaluated accordingly within the farm model.

Wind speed dependent power loss in yaw misalignment. The present baseline formulation models the power extraction of
a misaligned wind turbine using the cosine-law CP(γ) = CP cos(γ)pP, where CP is the power coefficient of the wind-aligned turbine, γ the misalignment angle with respect to the local flow direction, and pP the power loss exponent. Different values for the power loss exponent power loss exponents have been reported in the literature, ranging from the value of 1.4 found by Fleming et al. (2017), to 1.8 according to Schreiber et al. (2017), 1.9 for Gebraad et al. (2015), all the way to the ideal value of 3 that is expected if only the rotor-orthogonal ambient flow component contributes to power
extraction (Boersma et al., 2017). In addition, pP might also depend on the regulation strategy used by the on-board turbine controller. Here, the turbine power coefficient in misaligned operation is augmented as

$$C_{\rm P} = C_{\rm P} \cos(\gamma + p_{\rm P0})^{p_{\rm P} + p_{\rm P,a}(V - V_{\rm rated}) + p_{\rm P,b}},\tag{8}$$

where  $C_{\rm P}$  is the power coefficient of the flow-aligned turbine (possibly reduced by shear effects, as argued above),  $p_{\rm P0}$  is the misalignment angle at which the turbine produces maximum power, while V and  $V_{\rm rated}$  are, respectively, the rotor effective and rated wind speeds. Finally,  $p_{\rm P}$  is the baseline exponent, while  $p_{\rm P,a}$  and  $p_{\rm P,b}$  are free parameters that model a linear wind speed dependency of the cosine law.

**2.3 Parameter identification method**

[revised manuscript text omitted]

$$\mathcal{L}(S|_{\boldsymbol{p}}) = \left((2\pi)^{m} \det(\boldsymbol{R}^{-1})\right)^{-N/2} \exp\left(-\frac{1}{2} \sum_{i=1}^{N} \boldsymbol{r_{i}}^{T} \boldsymbol{R} \boldsymbol{r_{i}}\right).$$
(12)

5 Maximizing  $\mathcal{L}$  (or minimizing its negative logarithm), a maximum likelihood estimate of the parameters can be obtained as

$$p_{\text{MLE}} = \underset{p}{\arg\min} J(p), \tag{13}$$

where  $J(\mathbf{p}) = -\ln(\mathcal{L}(S|_{\mathbf{p}}))$ . The measurement noise covariance matrix  $\mathbf{R}$  can be estimated under mild hypotheses as  $\mathbf{R} = \sum_{i=1}^{N} \mathbf{r}_{i}^{T} \mathbf{r}_{i}$ , yielding  $J(\mathbf{p}) = \det(\mathbf{R})$ , leading to an iteration between a solution at given covariance and a covariance update step (Jategaonkar, 2015). However, in this paper the measurement noise covariance matrix is estimated a priori and therefore assumed to be known.

10 The cost function becomes therefore

$$J(\boldsymbol{p}) = \frac{1}{2} \sum_{i=1}^{N} \boldsymbol{r_i}^T \boldsymbol{R}^{-1} \boldsymbol{r_i}.$$
(14)

To ensure reasonable and physically viable solutions, parameters can be forced to stay within predefined upper (subscript ub) and lower (subscript lb) bounds, by adding the corresponding inequality constraints  $p_{lb} \leq p \leq p_{ub}$  to problem (13). As the parameter values and constraints can differ in magnitude, it is a good practice to scale all parameters such that a value of

15 1 corresponds to the upper bound  $p_{ub}$  and a value of -1 to the lower one  $p_{lb}$ . The optimization problem can finally be solved numerically by a suitable algorithm, such as Sequential Quadratic Programming (SQP) (Nocedal and Wright, 2006).

**2.0.2 Identifiability of parameters**

The Fisher information matrix  $F \in \mathbb{R}^{n \times n}$  is defined as

$$F = \sum_{i=1}^{N} \left[ \frac{\partial y_i}{\partial p} \right]^T R^{-1} \left[ \frac{\partial y_i}{\partial p} \right],$$
(15)

20 and describes the curvature of the likelihood function. It can be shown (Jategaonkar, 2015) that a lower bound (termed Cramér-Rao bound) of the covariance of the estimated parameter is given by

$$\boldsymbol{F}^{-1} = \boldsymbol{P} \leq \operatorname{Var}(\boldsymbol{p}_{\mathrm{MLE}} - \boldsymbol{p}_{\mathrm{true}}),\tag{16}$$

where  $p_{true}$  are the true but unknown parameters. The k-th diagonal element of P is a lower bound on the variance of the k-th estimated parameter, while the correlation between different parameters is captured by the off-diagonal terms of that same

25 matrix. The correlation coefficient between two parameters i and j is defined as

$$\Psi_{p_i,p_j} = \frac{P_{i,j}}{\sqrt{P_{i,i}P_{j,j}}},\tag{17}$$

where  $P_{i,j}$  denotes the *i*, *j*-th element (row, column) of **P**. By analyzing the estimated parameter variance, as well as the correlation between parameters, valuable insight into the well-posedness of the parameter identification problem can be readily obtained.

**2.0.3 Problem transformation and untangling using the SVD**

25

5 When some parameters are highly correlated or have large variance, the problem is ill-posed: it might exhibit sluggish convergence, or not converge at all, and small changes in the inputs may lead to large changes in the estimates. Such situations are difficult to solve in the physical space, because parameters are typically coupled together to some degree through the model.

To untangle the parameters, one may resort to the SVD (Golub and van Loan, 2013). By this approach (Hansen, 1987; Waiboer, 2007; Bo

10 , the original parameters are mapped into a new set of uncorrelated (orthogonal) parameters. Since the new unknowns are uncorrelated, one can set a threshold to their variance by using the Cramér-Rao bound, and only retain in the optimization those that are observable within the given data set.

The Fisher matrix F is first factorized as  $F = M^T M$ , where  $M \in \mathbb{R}^{Nm \times n}$  is defined as

$$M = \begin{bmatrix} \mathbf{R}^{-1/2} \frac{\partial \mathbf{y}_1}{\partial \mathbf{p}} \\ \mathbf{R}^{-1/2} \frac{\partial \mathbf{y}_2}{\partial \mathbf{p}} \\ \cdots \\ \mathbf{R}^{-1/2} \frac{\partial \mathbf{y}_N}{\partial \mathbf{p}} \end{bmatrix}.$$
(18)

15 Assuming a larger number of measurements than parameters (Nm > n), matrix M can be decomposed into

$$\underline{M} = \underline{U}\underline{\Sigma}\underline{V}^{T},\tag{19}$$

where  $U \in \mathbb{R}^{Nm \times Nm}$  and  $V \in \mathbb{R}^{n \times n}$  are the matrices of left and right, respectively, singular vectors, while

$$\Sigma = \begin{bmatrix} S \\ 0 \end{bmatrix}, \tag{20}$$

where  $S \in \mathbb{R}^{n \times n}$  is a diagonal matrix, whose entries  $s_i$  are the singular values sorted in descending order.

20 By using Eq. (19) and the factorization of F, the inverse of the Fisher information matrix can be written as

$$\underline{P} = \underline{V}\underline{S}^{-2}\underline{V}^{T}.$$
(21)

Note that the columns of the orthogonal matrix V are also the eigenvectors of P and  $s_i^{-2}$  the corresponding eigenvalues. Furthermore, P and F are symmetric and, based on the spectral theorem, diagonalizable.

The physical parameters p can now be transformed into a new set of orthogonal parameters  $\Theta$  by a rotation performed with the right singular values:

$$\Theta = V^T p.$$
(22)

For the transformed set of parameters, the Cramér-Rao bound on the variance of the estimates is the diagonal matrix  $S^{-2} \leq Var(\Theta_{MLE} - \Theta_{MLE})$ Therefore, a small singular value  $s_i$  corresponds to a large uncertainty in the corresponding orthogonal parameter estimation. To remove parameters that cannot be estimated with sufficient accuracy, matrix S can be partitioned as

$$S = \begin{bmatrix} S_{\rm ID} & \mathbf{0} \\ \mathbf{0} & S_{\rm NID} \end{bmatrix},\tag{23}$$

5 where  $S_{\text{ID}}$  contains the identifiable singular values, i.e. those such that  $s_i^{-2}